# Snow Depth Estimation on Lead-less Landfast ice using Cryo2Ice satellite observations

Monojit Saha[1], Julienne Stroeve[1,2], Dustin Isleifson[1], John Yackel[3], Vishnu Nandan[1,3], Jack Landy[4], Hoi Ming Lam[3]

1Centre for Earth Observation Science, Department of Environment and Geography, University of Manitoba, Winnipeg, Canada
2 Department of Earth Sciences, University College London. London, United Kingdom
3 Department of Geography, University of Calgary, Calgary, Canada
4 Centre for Integrated Remote Sensing and Forecasting for Arctic Operations (CIRFA), UiT The Arctic University of Norway, Tromsø, Norway

*Correspondence to:* Monojit Saha (saham1@myumanitoba.ca)

**Abstract.** Observations of snow on Arctic Sea ice are vitally important for sea ice thickness estimation, bio-physical processes and human-activities. While previous studies have combined CryoSat-2 and ICESat-2-derived freeboards to estimate snow depth over Arctic sea ice, these approaches require leads within the ice pack to estimate the freeboard heights above the sea surface. In regions such as the Canadian Arctic Archipelago (CAA), leads are scarce in winter, posing a significant challenge to estimate snow depth from altimeters. This study is the first assessment of the potential for near-coincident ICESat-2 and Cryosat-2 (Cryo2Ice) snow depth retrievals in a lead-less region of the CAA including validation with in-situ data. In lieu of sea surface height estimates from leads, snow depths are retrieved using the absolute difference in surface heights (ellipsoidal heights) from ICESat-2 and Cryosat-2 after applying an ocean tide correction based on tidal gauges between satellite passes on 29th April 2022. Both the absolute mean snow depths and distributions retrieved from Cryo2Ice were slightly underestimated (2 to 4 cm) when compared to in-situ measurements. All four in-situ sites had snow with saline basal layers and different levels of roughness/ridging which significantly impacts the accuracy of the Cryo2Ice snow depth retrievals. Differences between Cryo2Ice and in-situ snow depth distributions reflect the varying sampling resolutions of the sensors and the in-situ measurements. Cryo2Ice tends to miss snow depths greater than 30 cm, especially around ridges. The results suggest that it might be possible to estimate snow depth over landfast sea ice without leads. However, the observed biases of 2-4 cm likely stem from several factors: (1) discrepancies in sampling resolution between ICESat-2 and CryoSat-2, (2) the CryoSat-2 scattering horizon not aligning with the snow-ice interface due to snow salinity, density, and surface roughness, (3) the choice of retracker, and (4) potential errors in the altimeter's tidal corrections. Further investigation is needed to address these issues. Moreover, the proposed methodology for getting snow depth over lead-less landfast sea ice needs to be validated using in-situ datasets in other landfast sea ice regions in the Arctic.

## 1 Introduction

Changes in Arctic sea ice are affecting climate, ecosystems and traditional ways of living and harvesting (Meier and Stroeve, 2022). A critical component of the sea ice cover is its overlying snow cover, which has been challenging to accurately measure by satellites (Webster et al., 2018). Snow acts as an insulator, impacting both the growth and decay of sea ice (Maykut and Untersteiner, 1971). Snow also (1) limits the amount of light penetrating through the sea ice, affecting the timing of sea ice algae growth (Mundy et al., 2005); (2) contributes to the amount of freshwater discharged to the ocean, affecting its budget (Andersen et al., 2019); and (3) affects the heat exchange between the atmosphere and the sea ice (Andreas et al., 2005).

Using monthly composites of airborne laser and radar altimeter data collected during the Laser-Radar Altimetry (LaRA) mission over sea ice around Svalbard, Leuschen et al., 2008, suggested snow depth could be retrieved by differencing freeboards, though there was a lack of in-situ ground truth to validate results. Following this, studies have differenced coincident satellite radar (CryoSat-2; hereafter CS2) and laser (ICESat-2; hereafter IS2) altimeter freeboards to estimate pan-Arctic (e.g. Kwok and Markus, 2018; Kwok et al., 2020) and Antarctic snow depth (Kacimi and Kwok, 2020). However, significant uncertainties remain related to (1) differences in electromagnetic frequencies and spatial resolution (Fons et al., 2021); (2) whether or not the CS2 Ku-band radar returns originate from the snow/ice interface, which has been contested even for a dry and cold (below freezing) snow pack (Willatt et al., 2023, 2011; Nandan et al., 2017; de Rijke Thomas et al., 2023); (3) the influence of surface roughness over different length scales on the laser and radar waveforms (Landy et al., 2019); and (4) spatial heterogeneity of snow distributed over sea ice.

Earlier studies also faced challenges of having different orbits for CS2 and IS2, limiting the number of crossover points (Kwok & Markus, 2018). Kwok and Markus (2018) made a case for adjusting the CS2 orbit to achieve more overlaps with IS2, thereby improving both spatial and temporal coincidence. As part of the Cryo2Ice campaign, the CS2 orbit was raised by ~ 900 meters in August 2020 to significantly increase the amount of crossovers with IS2 (ESA, 2020). This realignment means that once in every 19 CS2 (20 IS2) cycles, the two ground tracks nearly align for hundreds of kilometers over the Arctic providing new opportunities to improve and validate snow depths retrieved by combining laser and radar freeboards.

Fredensborg Hansen et al. (2024) took advantage of the Cryo2Ice campaign to retrieve along-track snow depths along 7-km segments. In their study, they compared the derived Cryo2Ice snow depths over against snow depth products from passive microwave, snow models or climatologies and found uncertainties of 10-11cm.

T  his study is the first comparison of Cryo2ice snow depths to in-situ snow depth retrievals over landfast ice, evaluating retrievals along 300-meter and 1-km segments. This study also provides the first high-resolution in-situ validation of snow depths retrieved along coincident Cryo2Ice tracks near Cambridge Bay, Nunavut in the Canadian Arctic Archipelago (CAA). The CAA is a region with significantly different bathymetry and icescape than the Central Arctic (Galley et al., 2012). Sea ice in the CAA is landfast ice for the majority of the year (6 to 8 months) (Melling, 2002), and exhibits minimal ice drift (Galley et al., 2012), making it easier to match up IS2 and CS2 tracks. On the other hand, the tidal amplitudes within the shallow bathymetry of the CAA are larger than in the open ocean; posing an additional challenge compared to validation

studies in the Central Arctic Ocean. However, the most prominent challenge pertains to the lack of open water for estimating the local sea surface height (SSH) needed to reference the freeboards. Landfast ice grows along the narrow channels in the CAA and often lacks leads for several hundred kilometers (Galley et al., 2012). Therefore, assuming IS2 and CS2 are viewing the same landfast ice, the variation in SSH due to tidal variations must be known and corrected for between the two sensors. Our objective is to develop an approach to combine IS2 and CS2 along-track data in regions where the local SSH estimate is not readily available from satellite observations. The along-track Cryo2Ice retrieved snow depths are then validated using near-coincident in-situ snow depth observations. We further use in-situ snow property observations and satellite estimates of the surface roughness to examine the drivers of CS2 and IS2 height variability. Finally, the sources of bias in the retrieval process and major challenges are discussed.

## 2 Data and Methods

### 2.1 ICESat-2 (IS2)

The Advanced Topographic Laser Altimeter System (ATLAS) is the photon counting LiDAR system onboard ICESat-2. ATLAS emits low-energy 532 nm (green) pulses in three two-beam pairs which have a cross track spacing of 3.3 km between each pair with intra-pair spacing of 90 meters. The laser has a footprint size of 11 meters (Magruder et al., 2020). Detailed specifications can be found in Neumann et al., (2019).

In this study, the uncorrected ATL07 Sea Ice Height Release Version 6 available from the National Snow and Ice Data Centre (https://nsidc.org/data/atl07ql/versions/6#anchor-2) which are computed directly from ATL03 photon heights are used. ATL07 contains sea surface and sea ice heights derived from ATL03 photon heights that were aggregated into segment lengths consisting of 150 photons, resulting in variable along-track lengths over which these photos are accumulated. In the uncorrected ATL07 product, sea ice heights within the 25 km land-buffer are included despite low confidence in the geophysical corrections close to land (Kwok et al., 2023). The IS2 strong beam (gt2l) (referred to as IS2 2l) from ATL07 is used after assessing all three strong beams. The IS2 2l was ~1500 metre from the CS2 point of closest approach whereas beams 1l and 3l were ~2200 metre and ~4500 metre away, respectively.

The geophysical corrections applied to the ATL07 data are summarized in Table A1. Each correction is time-varying and has different impacts on the retrieved IS2 heights. Ocean tide corrections are provided every hour and can vary from -62 cm to +62 cm; the largest among the different geophysical corrections applied. The ocean tide corrections are obtained from the Global Ocean Tide Model 4.8 (GOT 4.8) (Kwok et al., 2021), which provides tidal predictions for all regions of the globe based on the assimilation of data from satellite altimetry and tide gauge measurements into a tidal model.

### 2.2 CryoSat-2 (CS2)

The SAR Interferometric Radar Altimeter (SIRAL) is the primary instrument on board CryoSat-2, which is a combination of a pulse-limited radar altimeter along with a Synthetic Aperture Radar (SAR) Interferometer system (SARIn). SIRAL operates

at Ku-band (13.575 GHz) and in three different modes with along-track sampling resolution of around 300 m and across- track resolution of 1600 m (ESA, 2013). Cryosat-2 operated in the SARIn mode in the CAA during the study period. Here we use the CS2 Level 2 Baseline E products available through the European Space Agency's EO-CAT web explorer (https://eocat.esa.int/). The CS2 Level 2 sea ice heights are re-tracked using the University College London (UCL) retracker (Tilling et al., 2018) which assumes a threshold (70%) on the first peak for diffuse echoes representing the mean elevation of the snow/sea ice interface within the footprint. This fixed threshold retracker is used in the CS2 Baseline E level product over sea ice floes in the SAR/SARIn mode.

Tidal corrections (ocean, long-period equilibrium, ocean loading, solid earth and geocentric polar) are included in the Level 2 Baseline E Cryosat-2 SAR/SARIn product (Table B2). The ocean tide, long-period equilibrium tide and ocean loading tide corrections used are retrieved from the Finite Element Solution 2004 Ocean Tide Model (FES 2004) (Cryosat-2 Product Handbook). The ocean tide corrections typically range from ± 50 cm.

**2.3 Sentinel-1 SAR**

Synthetic Aperture Radar (SAR) imagery from European Space Agency's Sentinel-1 satellite was used in this study in conjunction with IS2 and CS2. Sentinel-1 provides C-band dual-polarization SAR data which is available through the Google Earth Engine platform (https://developers.google.com/earth-engine/datasets/catalog/COPERNICUS_S1_GRD). The Sentinel-1 GRD files had already been pre-processed with the following corrections: GRD border noise removal, thermal noise removal, radiometric calibration and terrain correction. For this study, the cross-polarized VH backscatter was obtained from May 5th, 2022. The backscatter values were then converted to dB.

**2.4 Field Measurements**

The study site comprised a 75 km long NNE-to-SSW transect across Dease Strait (69°26'58.02"N 106°41'57.25"W to 68°46'42.48"N 106°55'52.10"W) (Figure 1), ~70 km west of Cambridge Bay, NU. This region connects Coronation Gulf and Queen Maud Gulf of the Kitikmeot Sea and is a part of the southern route of the Northwest Passage (Xu et al., 2021). Dease Strait is relatively shallow (maximum depth ~ 100 meters), and its narrow channel is covered by landfast ice normally between November and mid-July (Galley et al., 2012). CS2 and IS2 coincident tracks were identified using the CS2 and IS2 Coincident Data Explorer (https://cs2eo.org/) (Ewart et al., 2022). The tracks were ~1.5 km apart and passing by within 77 minutes of each other (Figure 1).

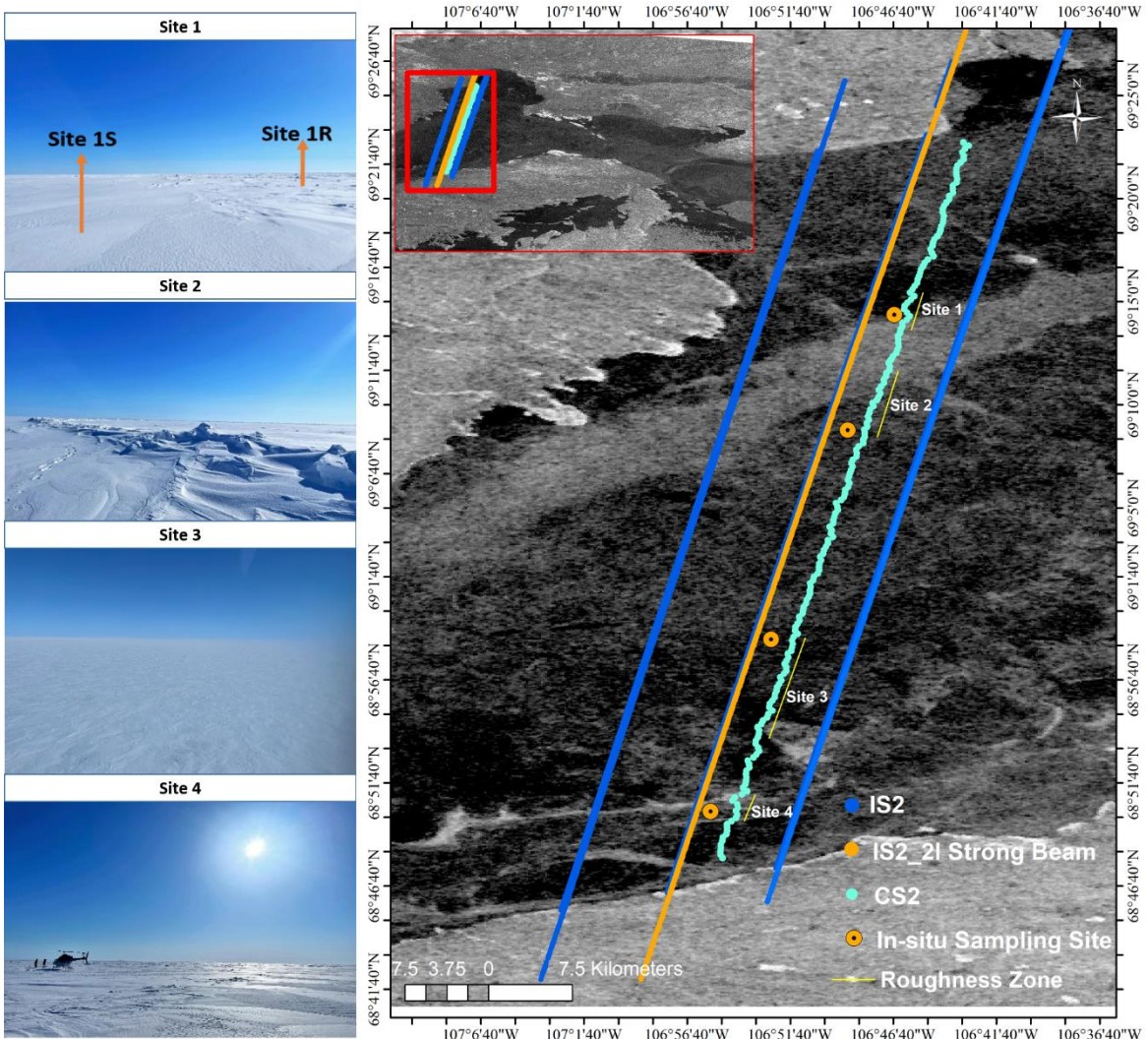

123

**Figure 1 Map shows the Cryosat-2 Points of Closest Approach (POCA) locations, IS2 2l Strong Beam and other IS2 beam, in-situ sampling locations and identified roughness zones. The background contains Sentinel-1 HH-pol SAR imagery. Site photos show the variation in snow roughness.**

In-situ snow depths were collected at four different sites (Sites 1-4) ranging from smooth, rough and mixed sea ice roughness zones. The transects were set considering wind direction as well as the sea ice surface features for each site. The sampling strategy was to ensure coverage of the Cryo2Ice along-track and across-track directions, taking into consideration the prevailing wind direction and different representative roughness features. At Site 1, two L-shaped transects representing the rough and smooth sea ice zones were conducted (Figure D1 (a). For Site 2, two different L-shaped transects were conducted to sample both the ridged ice areas as well as the smoother ice further away from the ridges (Figure D1(b)). For Sites 3 and 4 which had wider regions of smooth and rough sea ice respectively, two L-shaped transects were conducted (Figure D1 (c) & (d)). Based on Sentinel-1 SAR and field reconnaissance, Site 1 was classified as a rough and smooth sea ice transition zone;

Site 2 was a thin snow zone with significant ridging; Site 3 was a smooth sea ice zone with extensive areas of thin snow; and Site 4 was a rough sea ice site with extensive areas of thick snow. All sites were located equidistant between the IS2 strong beam and CS2 track to ensure the highest likelihood that snow depth sampling was representative of both sensors. The snow depth sampling direction was determined according to distinctive roughness features at individual sites, ensuring sufficient sampling distance in both the along- and across-track directions, representative of the prevailing east-southeast wind direction (ECCC, 2022) and snow dune pattern (Moon et al., 2019). Snow depth was surveyed using Snow-Hydro's automated snow depth magnaprobe, which has an accuracy of $\pm0.3$ cm on level sea ice and snow (Strum and Holmgren, 2018). The magnaprobe was reassembled and re-calibrated before each sampling effort to avoid instrument bias. Sampling was conducted by a single person to avoid variations in instrument handling and to maintain constant intervals between samples.

All four sites were surveyed on 01-05-2022 within 48 hours of the ICESat-2 and CryoSat-2 pass on 29-04-2022. The sites were accessed via helicopter and no sampling was conducted within 200 meters of the helicopter landing zone to avoid snow redistribution during landing. While the sampling interval was initially set at 5 m intervals to ensure spatial heterogeneity and to avoid spatial autocorrelation of the sampled snow depth values following Iacozza and Barber (1999), the sampling interval ranged between 2 to 3.8 m during the field sampling for all sites. There was no precipitation recorded during the sampling period, nor during the time interval between the CS2 and IS2 overpasses. Furthermore, high pressure dominated the region between 26-04-2022 and 04-05-2022 (ECCC, 2022) causing light surface winds. As such, snow redistribution between CS2 and IS2 overpasses and in-situ sampling was negligible. The air temperature varied between -11.7°C and -14.1°C during the sampling as measured at the Cambridge Bay, land-based meteorological station.

Snow geophysical properties including snow salinity and density were sampled from all four sites. Snow temperature was not measured because the temperature probe would not calibrate quickly enough between the short helicopter landing durations. For Site 1, two pits were sampled, one for the rough sea ice (Site 1a) and one for the relatively smooth sea ice zone (Site 1b). Single pits were excavated at the other three sites. Snow density was measured using a 66 cm$^3$ ($2 \times 5.5 \times 6$ cm) density cutter at 2 cm intervals and weighed in the lab. After, weighed samples were melted at room temperature for snow salinity measurement using a Cole-Parmer C100 Conductivity Meter (accuracy of $\pm 0.5\%$). Sea ice thickness and freeboard at each site was measured using a freeboard tape to an accuracy of 0.5 cm.

## 2.5 Estimating Snow Depth from Cryosat-2 and ICESat-2

Kwok et al (2020) calculates snow depth (SD) as the difference between IS2-derived total freeboard (snow + ice) and CS2-derived radar freeboard (CS2). Freeboard heights are computed relative to the instantaneous sea surface height interpolated from sea surface measurements from along-track leads (Kwok et al., 2020; Ricker et al., 2014). The CS2 radar freeboard is additionally adjusted for reduced Ku-band propagation speed through snow. While this approach has been applied to the Cryo2Ice campaign within the central Arctic (Fredensborg Hansen et al., 2024), freeboards require accurate estimation of the sea surface height which is dependent on the availability of leads within a reasonable distance (10's of km) along both the IS2 and CS2 track. No leads were detected along the portion of the IS2 and CS2 tracks in our study area and therefore the sea

surface height could not be reliably estimated. Therefore, we modified the approach used in Kwok et al., (2020) to instead use
the absolute sea ice heights measured from IS2 ATL07 (h(IS2)) and CS2 (h(CS2)) referenced to the WGS84 ellipsoid to
estimate SD (Figure 3). SD can be calculated as the freeboard differences under the assumption that Ku-band penetrates to the
snow/ice interface
$SD = \frac{h_{IS2} - h_{CS2}}{\eta s}$,                                                                                               (1)
Where $\eta$s is the refractive index of Ku-band microwaves which compensates for the propagation delay through the snow pack
(Kwok et al., 2020). The refractive index is calculated using ($\eta$s=(1+0.51$\rho$s)^1.5 (Ulaby et al., 1986), where the in-situ bulk
snow density ($\rho$s) measured from the field is used. The average snow density from all four sites is used to compute snow depth
for the entire track (Figure 8) while snow densities from each site are used to compute SD from corresponding portions of the
Cryo2Ice track (Figure 5).

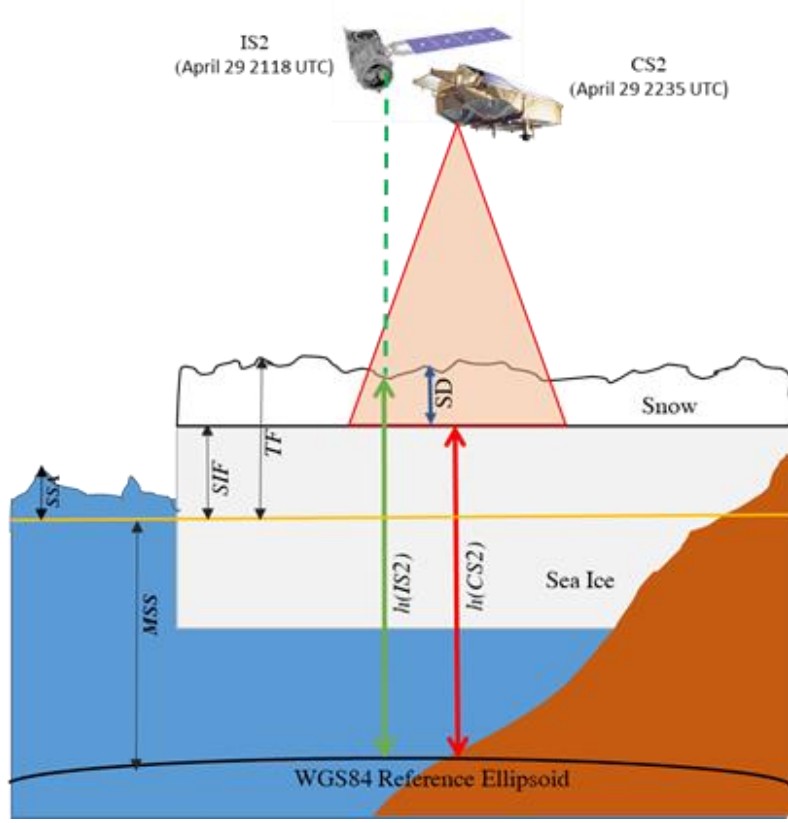


**Figure 2 Schematic showing the calculation of snow depth (SD) from ICESat-2 and Cryosat-2 over sea ice. The diagram illustrates**
**the representative heights for the sea surface anomaly (SSA), mean sea surface (MSS) in yellow, sea ice freeboard (SIF) and total**
**freeboard (TF). SD is shown with the blue arrow, IS2 surface height (*h(IS2)*) is shown with the green arrow and CS2 surface height**
**(*h(CS2)*) is represented by the red arrow. Land is orange.**

## 2.6 Data Processing

The uncorrected IS2 ATL07 heights ($h$ (IS2)) are referenced to the WGS84 ellipsoid which is also consistent with the CS2 heights (Figure 2). In our processing of the ATL07 data we apply the following geophysical corrections which are contained within the IS2 ATL07 product: ocean tide correction, long-period equilibrium tide and inverted barometer correction. We do not apply the mean sea surface (MSS) since it is based on decadal averages and therefore is not representative of the variation of sea surface heights within the 77 minute interval between the IS2 and CS2 passes. The geophysical corrections included within the CS2 data product are applied to the CS2 L2 sea ice heights. However, as mentioned previously the two products do not have the same tidal corrections.

Further, there is limited confidence in these individual geophysical corrections closer to land. The tides varied over a range of ~ 6.0 cm in Dease Strait in between the two passes based on the tide gauge data, so it was crucial to check if the tidal corrections contained within the products accurately accounted for tide differences in the ~77 minutes between passes. Therefore, after comparing the geophysical correction as explained in Section 2.6, an ocean tide correction factor is applied to the Cryo2Ice snow depths.

Since IS2 has a smaller footprint (Section 2.1 and 2.2), the IS2 ATL07 geolocated heights were averaged to be spatially congruent with the CS2 footprint giving snow depths estimates in the maximum along-track resolution of 300 m. Here, the IS2 photons are first averaged over 300 m length segments to match the along-track CS2 footprint and then co-registered based on the distance to the closest CS2 Point of Closest approach. Similarly, to reduce the impact of CS2 noise as explained later in Section 4.3 , the snow depths are also computed over 1-km. Therefore, each CS2 point is co-registered to the closest 300 metre ATL07 height segment. Snow depths computed from the IS2 and CS2 height differences were estimated following Equation (1), and subsequently adjusted with the ocean tidal correction. In order to      compare snow distributions representative of each sampled field site (S1 to S4), snow depth is compared over similar roughness zones.  Roughness zones corresponding to each Site are defined as a portion of the CS2/IS2 track which had IS2 surface roughness within one standard deviation of the IS2 derived surface roughness directly adjacent to the in-situ sampling site (Figure 1). The Cryo2Ice-derived snow depth corresponding to each roughness site was then compared against the in-situ snow distribution from the sampling sites.

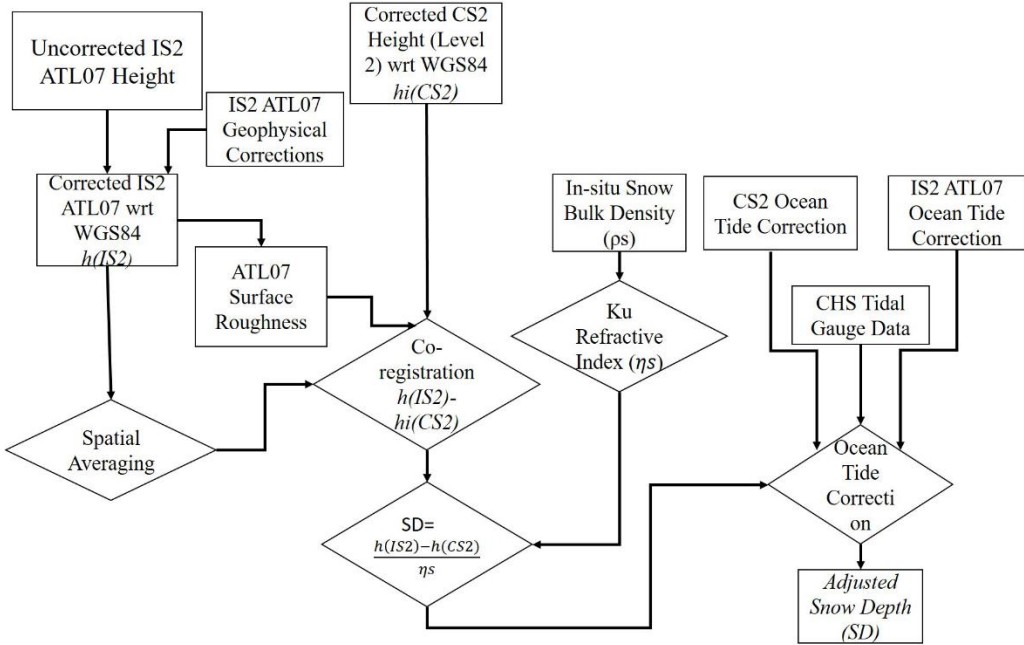

208

**Figure 3 Methodological workflow for retrieving snow depth (SD) from CS2/IS2 co-registered averaged ATL07 (*h (IS2)*) and Cryosat-2 heights (*h(CS2)*)) are subtracted following Equation 1. The differenced product is located at the Point of Closest Approach (POCA) of each CS2 footprint. The differenced product is then adjusted with the refractive index (ηs).**

**2.7 Adjusting for Sea Surface Height Variation**

Assuming IS2 and CS2 are viewing the same landfast ice, any variation in sea surface height over the short 77 minute interval between tracks is assumed to be due to tidal variations. The long-period equilibrium tide and ocean-tide with the inverted barometer corrections were compared between the sensors to identify differences between them. As mentioned earlier, different ocean tide corrections are applied to CS2 and IS2, with values ranging between +/-50 cm in CS2 and +/-62 cm in IS2 (Kwok et al, 2021, Cryosat-2 Product Handbook), and these have the most significant impact on the height retrievals (Figure C1, See Figure S1 in Bagnardi et al., 2021)) . Ideally, the ocean tide correction applied to IS2 and CS2 should account for the true variation in SSH due to local tides between the data acquisition passes. Although sea ice significantly dampens tides (Rotermund et al., 2021), tidal fluctuations, in this case the tidal corrections were found to be non-negligible. We compared the average ocean tide corrections to local tidal gauge predictions from the Canadian Hydrographic Service (CHS) (https://tides.gc.ca) which are based on real-time and historical tidal gauge measurements from the Cambridge Bay station. The CHS dataset provides instantaneous tidal variations at the CB station every 15 minutes with six observations between the IS2 and CS2 passes. The difference in ocean tidal corrections between the IS2 and CS2 pass was 7.9 cm on average along the track whereas the difference in water level was 6.0 cm according to the CHS data. The difference in height between IS2 and CS2 was therefore adjusted by a single value of 1.9 cm before the snow depths were computed (Figure 3) and this value then represents a systematic uncertainty on the final snow depth estimates.

## 2.8 Evaluating Other Sources of Uncertainties

One of the critical assumptions is that IS2 and CS2 tracks are roughly coincident i.e. both tracks are measuring roughly the same snow despite their reference ground tracks being ~1.5 km apart. To test this assumption, Sentinel-1 SAR VH backscatter was characterized across both the IS2 and CS2 reference ground tracks. The sentinel-1 backscatter is sensitive to surface roughness which roughly corresponds to the snow depths along-track (Cafarella et a;., 2019). Therefore, the Sentinel-1 backscatter which is used to compare the backscatter profiles along IS2 and CS2 tracks to determine if they are similar and therefore are seeing similar snow depth distributions. Given that IS2 has three different strong beams (IS2 1l,2l and 3l), we compare the SAR backscatter across all three tracks and compare it to the SAR backscatter along the CS2 track. We notice that along the IS2 2l track the SAR backscatter shows the most similar backscatter distribution as along the CS2 track (Figure 4(a)). This also aligned with the fact that the IS2 2l beam was the closest (~1.5 km) from the CS2 Points of Closest Approach (POCA) and therefore would see the most similar snow distributions. Therefore, the IS2 2l was considered for the subsequent Cryo2Ice snow depth calculations. The SAR pixels intersecting with the IS2 and CS2 track were used to calculate the mean backscatter along each track. The mean difference in SAR backscatter was -0.3 dB, less than 1 standard deviation of the backscatter of each track (Figure 4 (a)). Since both the tracks have similar backscatter, the assumption that they are coincident and observing snow packs with the same distribution is likely valid. Additionally, the difference in the point-to-point backscatter between IS2 and CS2 was also calculated to assess whether the difference in backscatter is consistent throughout the track (Figure G1). We see that the average difference in backscatter between the collocated points is within -+1 dB. The average difference in backscatter between IS2 and CS2 is 0.9 dB. Since both the tracks have similar backscatter, the assumption that they are coincident and observing snowpacks with the same distribution is likely to be valid in most cases.

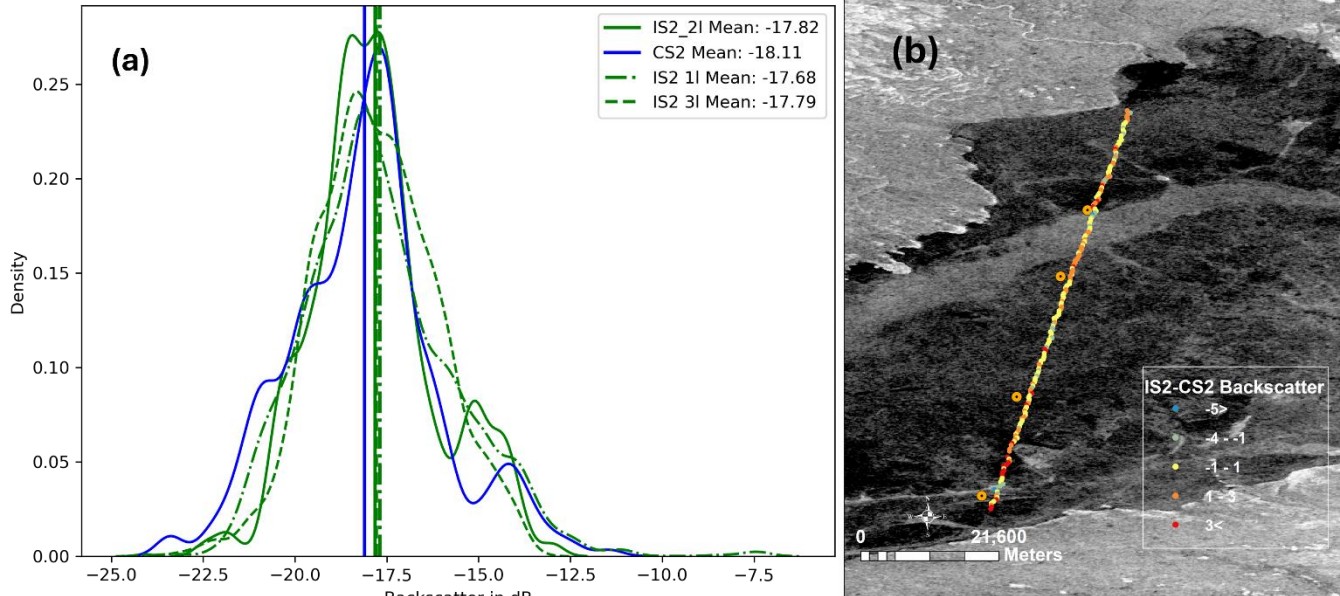


**Figure 4 (a) Sentinel-1 Backscatter in dB obtained from all the strong beams of IS2 (IS2 1l, 2l and 3l) and CS2 track locations. The**
**Sentinel-1 VH backscatter from 05-05-2022 is used for extracting backscatter along both the tracks to assess whether the observed**
**snow distribution is similar (b) Spatial d    istribution of the Sentinel-1 backscatter between IS2 and CS2 tracks, shown differences**
**in backscatter between IS2 and CS2 on     5th May 2023 .**
Landy et al (2019, 2020) demonstrated the importance of considering surface roughness in the radar data processing. Sea ice
surface roughness was computed across the IS2 track using the ATL07 sea ice height product.  Surface roughness was
calculated as the standard deviation of ATL07 sea ice height product following Farrell et al., (2020).  However, instead of the
25 km distance set  for pan-Arctic studies, the regional differences in surface roughness were calculated over 300-meter length
segments to maintain consistency with the spatially averaged ATL07 heights.
Previous studies measured or modelled the dominant scattering surface over first-year sea ice (FYI) at Ku-band (Nandan et
al., 2017, 2020; Willatt et al., 2011) several to many centimeters above the snow/sea ice interface even for cold snowpacks.
Nandan et al. (2017, 2020) argue that when brine is present within the snowpack, the dominant scattering horizon at Ku-band
is shifted upwards by approximately 7 cm above the snow/sea ice interface. Mallett et al., (2020) further demonstrated that the
use of fixed snow densities introduced significant biases in the snow depth retrievals. Provided snow salinity impacts the
location of the Ku-band dominant scattering horizon (Nandan et al., 2017), an assessment was conducted to test the bias
introduced by choosing different snow bulk densities by (a) assuming Ku- band microwaves penetrate completely through the
snow layers to the sea ice surface and (b) Ku-band microwaves penetrates through layers with snow salinity less than 1 ppt.
The corresponding average in-situ snow bulk densities from (a) the complete snow layer (b) snow layers with less than salinity
of 1 ppt were used to compute refractive indices followed by respective snow depth calculations. There was negligible
difference in the refractive index (<0.05) considering the snow bulk densities with difference in salinity and therefore the
average bulk densities from the complete snow pack was used in this study.
**3. Results**
**3.1 In-Situ Snow Depths and Distributions**
In-situ snow depths demonstrate significant spatial variability among the four sampled sites (Figure 5). The mean snow depth
from the four different sites varies between 9 and 17 cm, and all sites have positively skewed distributions (Figure 5). Site 2
also has some exceptionally high snow depths (> 90 cm), corresponding to the ridged areas (Figure 5) and therefore show
higher standard deviations (Figure 5).  Sites 2 and 3 have similar snow distributions (Figure 5) but the presence of ridging in
Site 2 results in a wider tail compared to Site 3. The maximum snow depth of 80 cm was recorded in Site 2 which was picked
up directly adjacent to the ridge. Site 4 has the highest mean snow depth (Figure 5) as well as the thickest tailed snow
distribution (Figure 5). The distinctive snow depth characteristics were also evident from the standard deviation of snow depth
among the four sites. Site 2 which had significant ridging also had the highest standard deviation of snow depth (15.8 cm).
Site 1R and Site 4 which had rougher sea ice both had high standard deviations of snow depth (13.7 (Site 1R) and 13.9 (Site

282    4)).

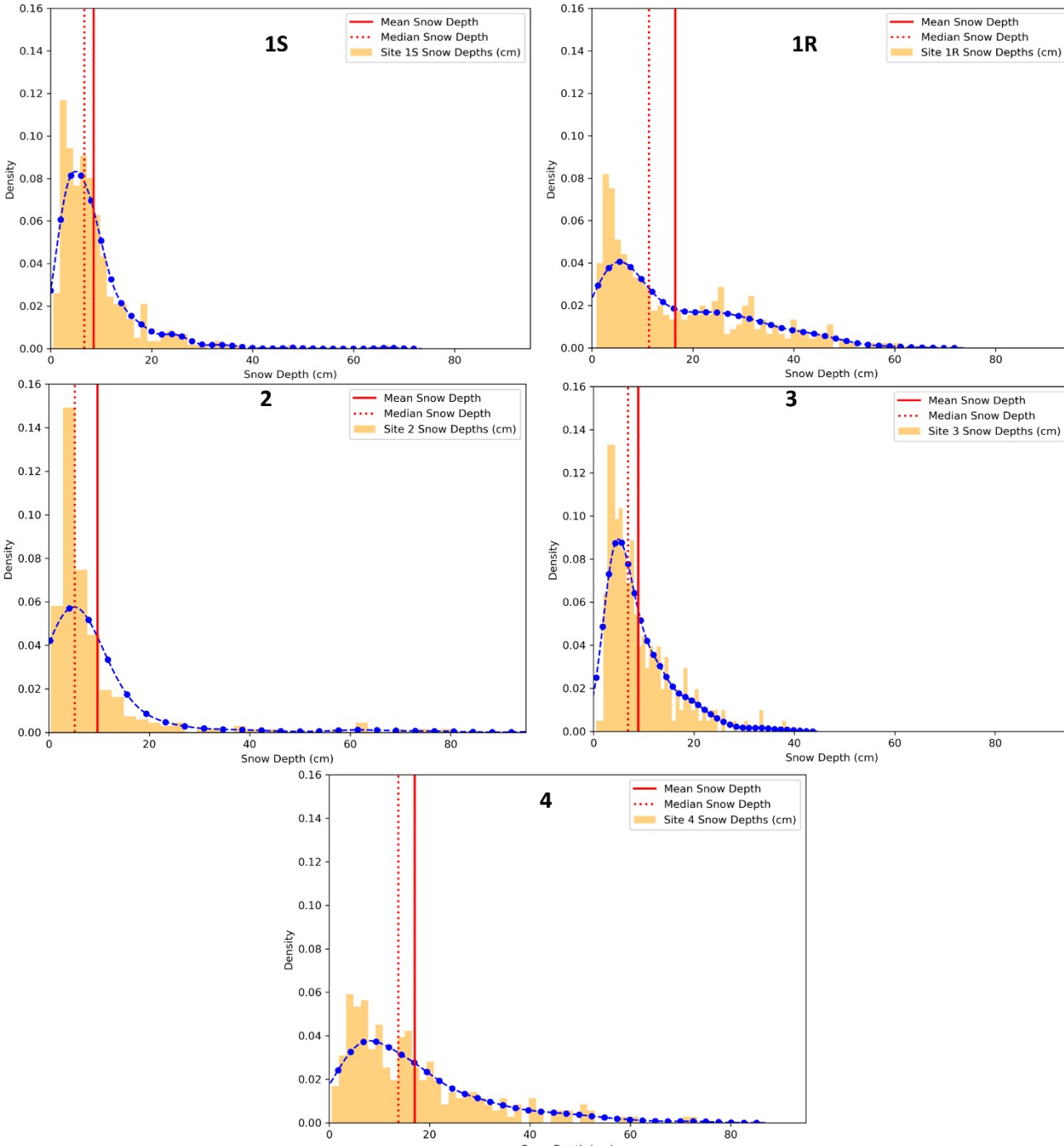


**Figure 5 Snow depth distributions from the four in-situ measurement sites along the Cryo2Ice transect. The density distribution**
**curve is shown in blue.**

**3.2 Snow Geophysical Parameters**
Mean snow salinity varies between 1.5 to 3.0 ppt for Sites 1S, 2, 3 and 4, whereas at Site 1S the snow salinity is 6.78 ppt
(Figure 6). The mean snow bulk density varies between 0.358 and 0.374 $g/cm^3$ in all sites except Site 3 where the mean snow
density is 0.248 $g/cm^3$.
Vertical profiles of snow salinity and bulk density present further insights. As shown in Figure 6, the snow density patterns
are similar for Sites 1R, 1S, 2 and 4 with bulk density ranging between 0.260 to 0.420 $g/cm^3$ and lower at the base of the
snowpack than the surface (Figure 6). The snow density varies in the different snow layers but there is a general trend towards
higher densities at 4 to 7 cm above the snow-ice interface at all sites (Figure 6). This is attributed to the presence of a wind
slab snow layer most prominent at Sites 1R, 2 and 4.
Snow salinity shows higher salinities closer to the snow-ice interface but decreasing with height up the interface (Figure 6 (a)).
For snow pits greater than 7.5 cm thick, the salinity is less than 1 ppt closer to the air-snow interface. There is a spike in salinity
between 5 to 3 cm from the snow-ice interface at Site 3 that corresponds to the high bulk density snow layer (Figure 6(b)).

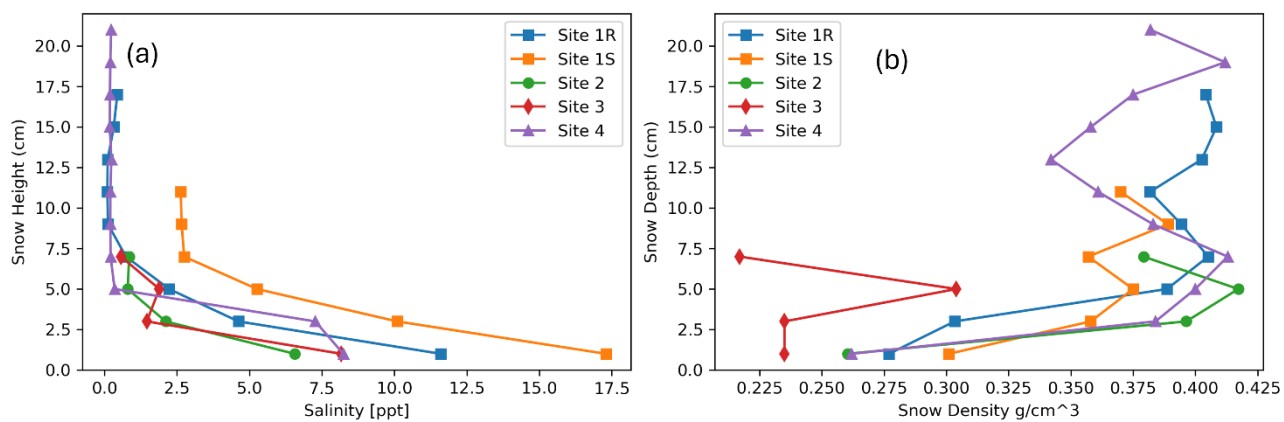


**Figure 6 (a) Snow salinity and (b) Snow density change by snow pack depth at the four snow sampling sites. Zero snow depth in**
**both plots represents the snow-ice interface.**
**3.3 ICESat-2/Cryosat-2 Derived Snow Depths**
Snow depths were calculated based on the ellipsoidal height difference between the IS2 2l and CS2 after adjusting for the
difference in tides as explained in Section 2.6 (Figure E1). IS2 2l was closest to the CS2 Points of Closest Approach (POCA)
which ensured that the uncertainty due to the difference in spatial colocation of IS2 and CS2 was minimized as explained in
Section 2.7. The CS2 ($h(CS2)$ and IS2 ($h(IS2)$)) heights show a general pattern of lower CS2 heights relative to co-registered
IS2 heights (Figure 7). The correlation of the CS2 ellipsoidal height with the Cryo2Ice snow depth (0.2509) is higher than the
IS2 ellipsoidal heights (-0.1213) which implies that the snow depths would be impacted more by the noise in CS2 heights
compared to IS2. The *h(IS2)-h(CS2)* differences range between -26.5 cm and 50.0 cm with a mean difference of 7.9 cm. 20%
of the calculated differences are negative which are distributed randomly along the track (Figure 8). While negative snow
depths don't have a physical basis, we include them in the subsequent snow depth calculations to not discard the impacts of
altimeter noise on the retrieved heights (Fredensborg Hansen et al., 2024). The noises in the CS2 heights as evident in Figure
7, corresponds with the large negative snow depth values (Figure 7, Figure 8). Therefore, to reduce the negative bias in snow
depths due to the CS2 noise, we exclude negative snow depth values which are two standard deviations away from the mean
Cryo2Ice snow depths in the subsequent calculations (Figure 9).
The adjusted mean snow depth across the whole Cryo2Ice track is 7.4 cm (Figure 5). A maximum snow depth of 39.4 cm is
retrieved from Cryo2Ice, at a length scale of 300 m which is significantly lower than the maximum snow depths measured in
situ > 90 cm.

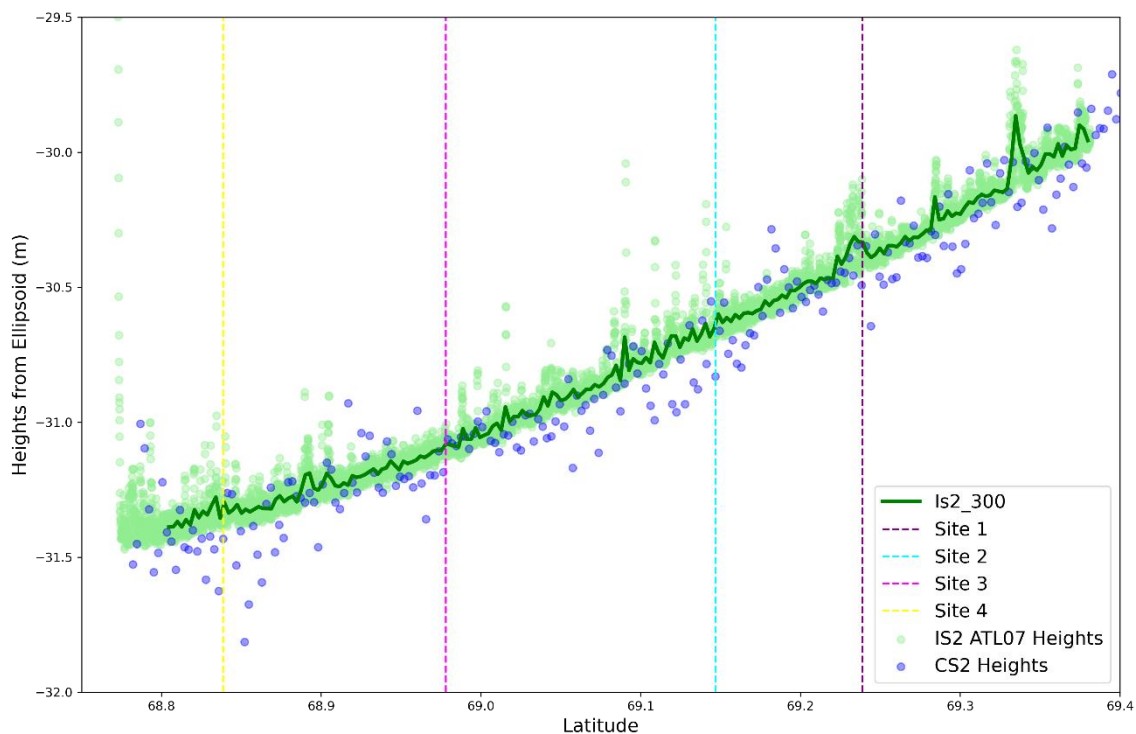



**Figure 7 IS2 ATL07 sea ice heights plotted along with CS2 surface heights. Note, the reported heights are relative heights and can**
**be negative because of the WGS84 ellipsoid reference heights in the study area. The light green color indicates the raw ATL07**
**heights (IS2 ATL07 Heights). The solid green line indicates the aggregated ATL07 heights aggregated every 300 meters (IS2_300).**
**The purple color indicates the CS2 Heights.**

Snow depths shown in Figure 9 display a right-skewed distribution with a sharper and heavier tail compared to a normal distribution. This is consistent with the distributions obtained from the in-situ snow sites (Figure 5). Analyzing the spatial distribution of the retrieved snow depths demonstrates that there is high spatial variability in the retrieved Cryo2Ice snow depths (Figure 8).

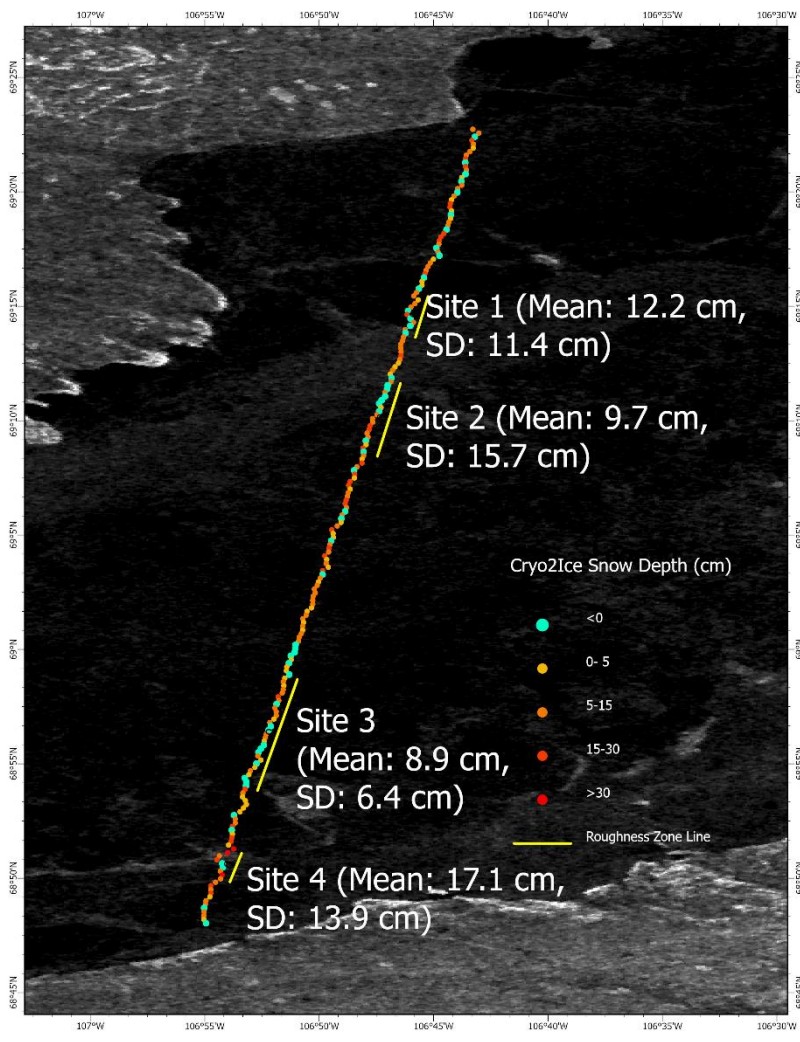

**Figure 8 Spatial distribution of 300-m scale Cryo2Ice snow depths across the CS2 and IS2 derived track. The background image is a Sentinel-1 HH backscatter image from 5-05-2022. The mean and standard deviation (SD) of the in-situ snow depths are labelled for surveyed sites included inside brackets.**

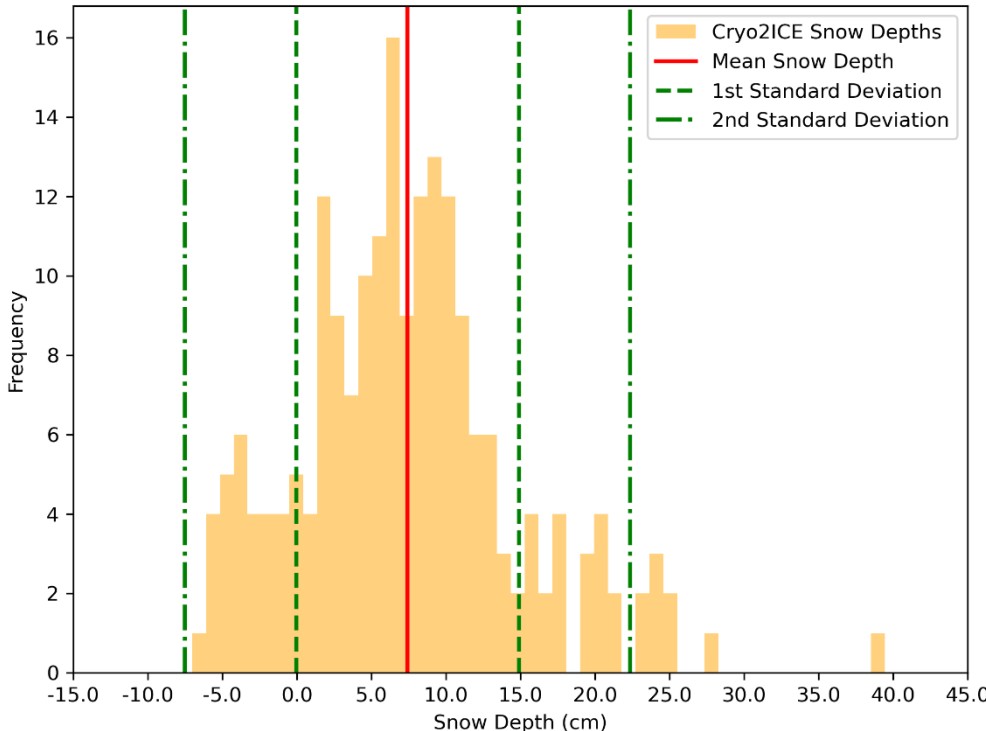

**Figure 9 Histogram showing the density distribution of the retrieved snow depth in the native 300 m resolution along the Cryo2Ice track with the mean and the median snow depths. Negative snow depths greater than 2 standard deviations from the mean snow depth were removed to reduce the impact of CS2 noise.**

## 4 Discussion

### 4.1 Snow Depth: Cryo2Ice vs In-situ

Previous field observations from Yackel et al. (2019) and Nandan et al. (2020) suggest that mean snow depth on FYI in Dease Strait during late winter ranges between 10 and 30 cm depth (Table 1). While our mean in-stu snow depth measurements (11.9 cm) within the typical range reported in previous surveys, we see that the Cryo2Ice mean snow depth (7.44 cm) underestimated the observed snow depths (Table 1).

**Table 1 In-situ snow depth measurements at Dease Strait. The range of mean snow depths represents the range of mean snow depths**
**retrieved from the sampled sites.**

| Sampling Period | Mean Snow Depth (cm) | Number of Sites Sampled | Total Number of Samples | Sampling Technique | Reference |
|---|---|---|---|---|---|
| 20 April to 9 June, 2014 | 13.5 | 24 | 24 | Snow Pits | Campbell et al., (2016) |
| 12 May to 17 June , 2014 | 20.8 | 2 | 60 | Meter Rule Sampling | Diaz et al., (2014) |
| 19-22 April, 2014 | 12.0/18.0 | 20 | 5200 | Meter Rule Sampling | Zheng et al., (2017) |
| 23-26 May, 2016 | 12.0/22.0 | 4 | 2100 | Meter Rule Sampling | Moon et al., (2019) |
| 01-08 April, 2017 | 17.0/ 35.0 | 5 | 2161 | Magnaprobe Sampling | Moon et al., (2019) |
| 17-19 May, 2018 | 20.9 / 21.8 | 3 | | Magnaprobe Sampling | Yackel et al., (2019) |
| 1 May, 2022 | 11.9 | 4 | 1596 | Magnaprobe Sampling | This Study |
| Cryo2Ice Snow Depths | 7.44 (Mean), 39.4(Maximum) | | | | |


Cryo2Ice snow depths showed similar relative patterns when compared to in-situ snow depth sampling. The thinnest (Site 3)
and thickest (Site 4) mean snow depths found in the in-situ measurements are corroborated with Cryo2Ice snow depths as well.
The Kruskal-Wallis non-parametric test (Kruskal &Wallis, 1952) was conducted to assess statistically significant differences
between the snow depths retrieved from the in-situ and Cryo2Ice. The test results show significant difference between in-situ
sites which was also evident in the corresponding Cryo2Ice snow depths.
Considering the median bias of snow depths reduces the impact of the outliers i.e. the retrieved negative snow depths as well,
Cryo2Ice snow depths are on average 3.07 cm thinner than the in-situ data, which is a 1 cm larger difference than the manual
tidal correction we applied to compare the CS2 and IS2 track heights (i.e., the largest known systematic uncertainty during
processing) (Figure F1). This pattern of a few cm mean snow depth underestimations by Cryo2Ice is consistently observed
across four sites (Figure 10)(Table F1). It is evident that while IS2 has a much finer resolution, the larger footprint of CS2
means that the spatial variability of snow depths under the kilometer scale are not well represented by Cryo2Ice. For instance,
the Cryo2Ice snow depths are consistently truncated at the thick end of the distribution, with at least some portion of the in-
situ distributions above ~30-50 cm seemingly unresolved from space (Figure 10).
We also notice that the Cryo2Ice snow depth distributions are generally wider than the in-situ distributions which is due to the
impact of the significant negative snow depths which are included in the calculation. These negative snow depths, while
included in the initial calculations to reflect the true native resolution results, don't have a physical basis, leading to artificial
widening of the distributions in Figure 10.

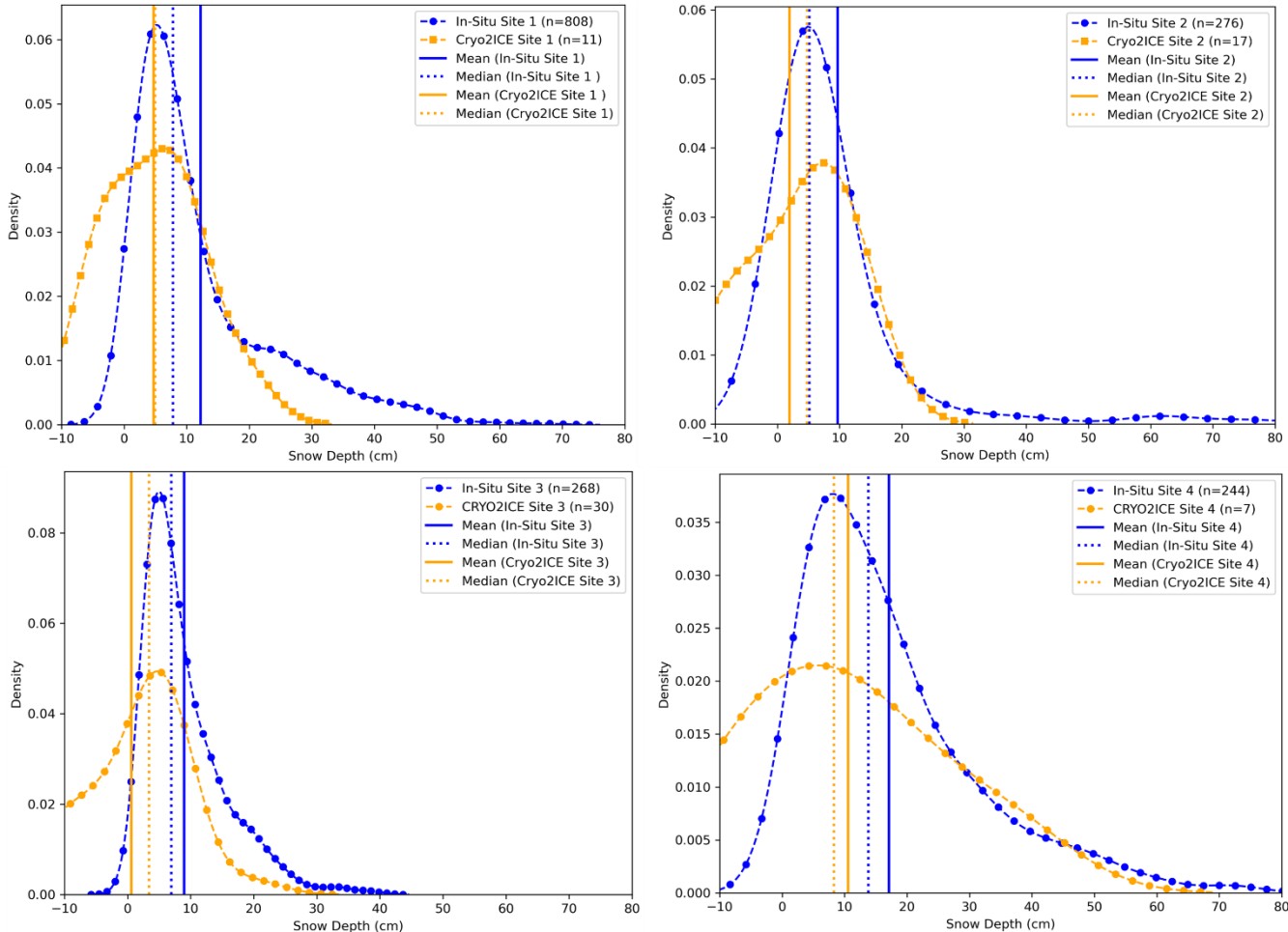


**Figure 10 Probability Density plots comparing In-Situ snow depths to Cryo2Ice retrieved snow depths along with the median and mean values. Different snow bulk densities were used to calculate the refractive index and subsequentyly Cryo2Ice snow depths for each site (Site 1-0.399 g/cm$^3$, Site 2- 0.398 g/cm$^3$, Site 3- 0.217 g/cm$^3$, Site 4-0.381 g/cm$^3$). The detailed statistics for the comparison are provided in Table F1.**

## 4.2 Adjusting for the Difference in CS2 and IS2 Footprint

As noted in Section 4.2, the difference in CS2 and IS2 footprint size with IS2 having a significantly smaller footprint compared to CS2 leads to a significant underestimation of the retrieved snow depths in the native 300 m resolution. Therefore, to reduce the impact of this artificial underestimation of the distribution, we average both IS2 and CS2 over a larger along-track distance. While averaging the CS2 and IS2 over 1-km causes some of the prominent roughness features such as ridges to be missed by Cryo2Ice, the snow depths from the 1-km CS2 and IS2 averaged heights are more realistic representations of the snow distributions when compared to in-situ (Figure 11). The average snow depth from the 1-km averaged CS2 and IS2 heights represents the overall shapes of the in-situ snow depths better compared to the native 300-meter averaged heights (Figure 11). The shapes of the distributions are well represented especially in Site 1 and 2. We also notice that shapes of the Cryo2Ice snow depth distributions match best in Site 1 and 2 compared to in-situ. However, the general underestimation of snow depths is reflected within most of the Sites (Site 1, 2, 3) except Site 4 which seems to overestimate the snow depth (Figure F2). The average snow depth retrieved from the 1-km averaged product is 7.80 cm which is slightly higher than the 300-meter averaged product presented in Section 3.3. The median bias between the in-situ and the 1 km averaged product is less than 2 cm in Sites 1 and 2. (Figure 11) (Tabel F2).

Comparing the shapes of the distributions, we see that almost all the sites have similar snow depth distributions compared to in-situ sites (Figure 11). However, a significant portion of the tails of the distributions are still missing which was also evident in the 300 m snow depth product. While the shapes of the distributions in Sites 3 and 4 are similar compared to in-situ, the peaks of the distribution don't coincide well. Cryo2Ice snow depths in Site 1 has the most similar distribution to in-situ compared to the other sites. In Site 2 we also see very similar snow depth distributions between Cryo2Ice and in-situ even between the 20 to 30 cm snow depths. While the shapes of the distributions match well in Site 3, we see a shift towards negative snow depths indicating that negative snow depths caused by noise in CS2 have larger impacts here in the smoother sea ice. Cryo2Ice seems to perform worst in Site 4 which is the roughest sea ice zone, with Cryo2Ice snow depths being overestimated when compared to in-situ. This is also evident in the shapes of the 1-km adjusted snow depth product which seems to be skewed towards higher snow depth values (Figure 11). Therefore, after adjusting for the difference in footprint size and averaging over 1-km along-track distance, the overall snow depth distributions are more similar to in-situ for the majority of the sites.

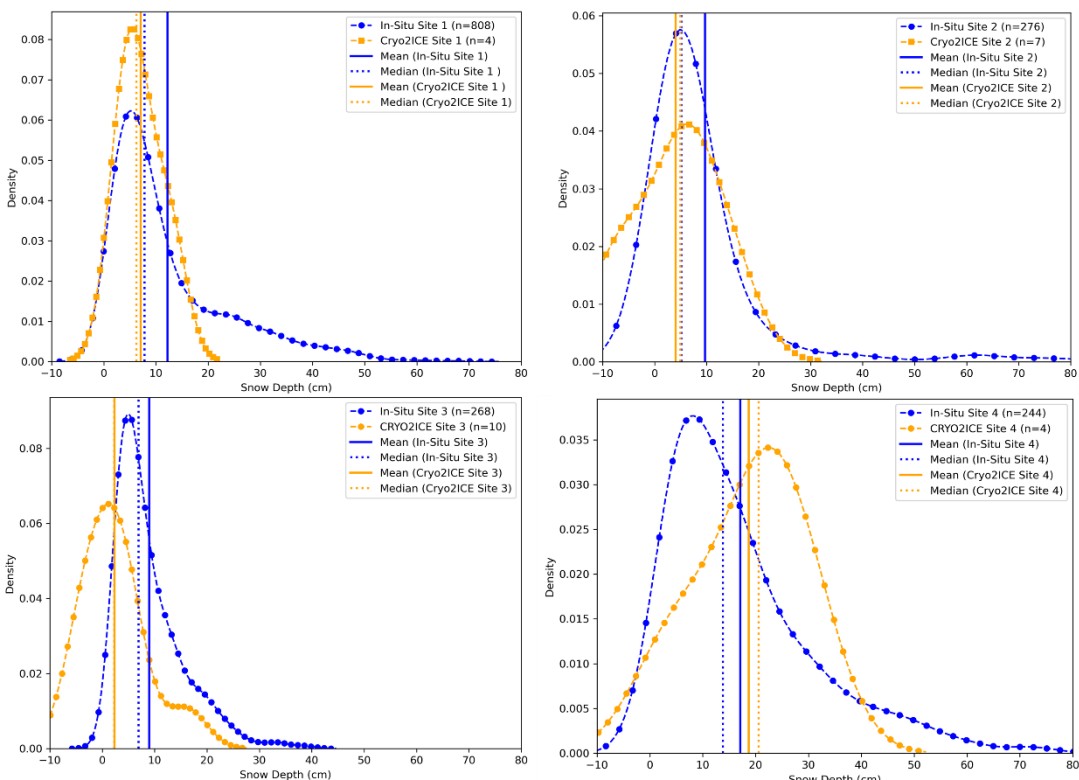

396

**Figure 11 Probability Density plots comparing In-Situ snow depths to Cryo2Ice retrieved snow depths retrieved from 1-km averaged CS2 and IS2 heights along with the median and mean snow depth values. Different snow bulk densities were used to calculate the refractive index and subsequentyly Cryo2Ice snow depths for each site (Site 1-0.399 g/cm$^3$, Site 2- 0.398 g/cm$^3$, Site 3- 0.217 g/cm$^3$, Site 4-0.381 g/cm$^3$). The detailed statistics for the comparison are provided in Table F2**

**4.3 Snow Geophysical Properties and Cryo2Ice Retrievals**

Both snow salinity and bulk density changes across the snowpack layer impacts the IS2 laser and CS2 radar waveform interactions with the snowpack. While the IS2 green laser is mostly impacted by the air-snow interface conditions, CS2 radar waveforms interact with different layers of the snowpack and the dominant scattering horizon and subsequently radar heights are impacted by the snow properties. There were significant differences among the snow salinity and density characteristics (Figure 6) between the surveyed sites. However, we notice that higher snow depths i.e. greater than 30 cm were picked up better in Site 4 which also had the lowest mean salinity with 17 cm out of the 22 cm deep snowpack being non-saline. Therefore, the maximum intensity of the CS2 backscatter may have been sourced from closer to the sea-ice interface in Site 4. On the contrary, highly saline layers can potentially raise the height of dominant scattering intensity of the Ku-band radar leading to overestimated CS2 heights ($h(CS2)$) and subsequently lower mean snow depth compared to in-situ values. This phenomenon of snow depth underestimation was evident in Sites 1 and 2 potentially because of the sharp increase in snow salinity within

the first 5 cm (from the air-snow interface) of the snowpack (Figure 6) and may have contributed to ~ 2 cm underestimation
of Cryo2Ice snow depths.
The impact of snow bulk density on the Cryo2Ice retrievals was less likely except for the presence of wind-slab layers which
are identified as stark increases in snow bulk densities within the snowpack.  The wind-slab layers were identified in Site 1R
where the density reached to 0.425 g/cm$^3$ compared to 0.358 to 0.374 g/cm$^3$ on average throughout the snowpack which may
have caused hindrance to Ku-band penetration which may have contributed to median underestimations. The presence of this
high-density snow layer along with the reduction in Ku-band speed due to power attenuation of Ku-band microwaves may
potentially cause a cumulative upward shift of the dominant scattering horizon resulting in underestimation of snow depths.
However, it is difficult to ascertain such uncertainties to a single physical factor due to interdependency of the processes.
**4.4 Sea Surface Height Estimation and Cryo2Ice Retrievals**
Canadian Hydrographic Service (CHS) tidal predictions for 29 April 2022 suggest the satellite overpasses occurred during a
low tide period. According to the predictions, the water level was 6 cm higher for the IS2 pass at 21:18 UTC than for the CS2
pass at 22:35 UTC (Figure C1). This 6 cm water level difference should ideally be accounted for by the difference in IS2 and
CS2 ocean tide corrections. The IS2 ATL07 heights were reduced by a mean ocean tide correction of -0.71 cm whereas the
CS2 Heights reduced by an average ocean tide correction of -8.64 cm. Therefore, the difference between IS2 heights and CS2
heights was increased by 7.9 cm due to the ocean tide correction adjustment but the CHS predictions suggest it should have
been only 6.0 cm. This 1.9 cm difference would introduce a 25.5 % bias in retrieved snow depths, given the approx. mean
snow depths we measured in-situ. This error could be attributed to the ocean tide corrections used in IS2 and CS2 originating
from two different models i.e. GOT 4.8 (IS2) and FES 2004 (CS2). To put this source of error into wider context, past CS2
and IS2 coincident tracks from 15-04-2021 and 14-05-2021 were also analysed. We found a bias of 2 to 5 cm when compared
with the CHS dataset, meaning that we can expect ~15-40% systematic uncertainty in Cryo2Ice retrieved snow depths owing
to the uncertainty in tidal differences between satellite passes. This is a significant uncertainty, but it is systematic and varies
at the length-scale of the tidal corrections (100s km), so it will not affect the *relative* variations in retrieved snow depth along
track, only their *absolute* magnitude. Therefore, Cryo2Ice seems capable of measuring the relative variations in snow depth
between different locations of the CAA without the availability of sea surface reference tie-points.
**4.5 Surface Roughness and Cryo2Ice retrievals**
Surface roughness calculated from IS2 was used to analyze the Cryo2Ice snow depths between sites with different roughness.
There was only a weak positive correlation (R$^2$ 0.04) between surface roughness retrieved from IS2 and Cryo2Ice snow depths.
Site 4 had the highest mean surface roughness (4.58 cm) whereas the other sites had roughness ranging between 2.4-2.7 cm.
Although there was significant ridging in Site 2 and IS2 does pick up some of the ridges (Figure 7), the mean surface roughness
is low (2.48 cm) because of the extensive areas of thin snow cover which dominates the laser returns. Site 4 had the highest
snow depth as well as highest surface roughness from IS2 which also corresponds with the highest median bias (Table F2).
Significant variation in surface type in Site 4 is also evident from the large variation retrieved backscatter from Sentinel-1 (-
5dB to 3dB)(Figure 4(b)) which was not very well represented from the snow depth estimations from Cryo2Ice. Therefore, we
notice that Cryo2Ice performs poorly in regions with relatively high surface roughness. The presence of isolated ridges and
the deeper snow accumulated around them may have been missed by the CryoSat-2 radar given the larger impact of level ice
versus ridges on the backscattered power which may explain the underestimation in Sites 1 and 2. The ridge heights may also
be underestimated with current ICESat-2 processing methods (Ricker et al., 2023) meaning that snow depths would be
underestimated. The surface roughness from IS2 computed and compared well to the roughness features picked up from the
snow depth variations with higher roughness zones having higher snow depths from Cryo2Ice e.g. Site 4. However, the
difference in spatial resolutions between IS2 and snow depths from Cryo2ice means that finer scale surface roughness features
were missed by Cryo2Ice especially in the 1-km averaged snow depth product.

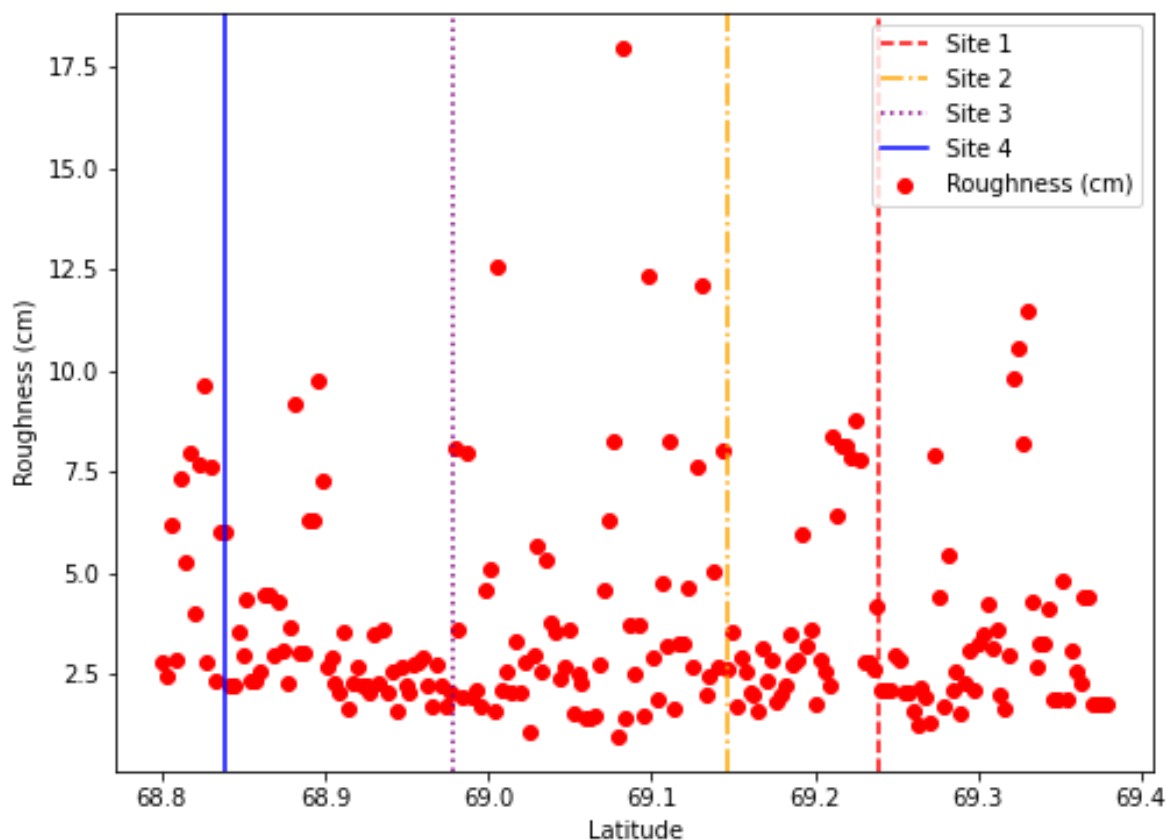

**Figure 12 Variation in surface roughness along the Cryo2Ice track at the four in-situ snow thickness validation sites**

**5 Conclusion**

Accurate snow depth monitoring over landfast ice in the Canadian Arctic Archipelago (CAA) is important for communities
that rely on landfast ice for transportation and their livelihood (Mahoney et al., 2009). It is imperative to monitor snow depth
in the CAA as there have been reports of declining snow depths at a rate of 0.8 cm per decade in Cambridge Bay and at other
locations in the CAA (Howell et al., 2016; Lam et al., 2023). Moreover, the reported snow depth on sea ice trends were highly
correlated to the declining sea ice thickness. Therefore, this study explores the potential of retrieving snow depth using
Cryo2Ice in a lead-less regions of the Canadian Arctic Archipelago.        Snow depth from Cryo2Ice is retrieved based on the
elevation difference between IS2 and CS2 sea ice heights from a common ellipsoid as opposed to the popular freeboard
differencing method. The instantaneous difference in sea level between the        77-minute difference between the CS2 and IS2
passes is accounted for by adjusting the ocean tide corrections with local tide model predictions. The snow depths retrieved
from Cryo2Ice compare favourably with in-situ snow depth measurements when averaged over 1-km segments of the tracks.
The relative snow depth patterns from in-situ field sites were corroborated with Cryo2Ice measurements, i.e. the thinnest and
thickest snow depth regions were picked up correctly by Cryo2Ice. The 300 meter averaged Cryo2Ice snow depths shows an
average of 7.44 cm which is slightly underestimated when compared to in-situ measurements from this study (11.9 cm  and
previous studies conducted at the Dease Strait. While the ~2 to 3 cm underestimation demonstrates that Cryo2Ice can estimate
snow depth with reasonable accuracy after adjusting for the tidal uncertainty (Fre    densborg      Hansen et al., (2024) reports
uncertainties of 10-11 cm uncertainties), there are still significant sources of both systematic and random uncertainties that
need to be addressed. We note that median biases ranging from 2 to 5.5 cm are reported among the different Sites which is
often higher than the tidal correction applied (1.9 cm).

The site-wise comparison between in-situ snow depths and Cryo2Ice snow depths show that Cryo2Ice performs well in regions
with moderately thin and smooth snow on sea ice i.e. ranging between 5 to 20 cm while it struggles to pick up snow depths
greater than 30 cm irrespective of the roughness characteristics. This phenomenon is largely attributed to the difference in
footprint size between CS2 and IS2 where the large footprint of CS2 missed a lot of the high snow depth sites particularly the
ones close to the ridges which are otherwise picked up by IS2. We also notice that negative snow depths mostly retrieved from
rougher sea ice zones spatially coincides with the noisy CS2 heights which are significantly higher than the IS2 heights. These
negative snow depths (20 % of the Cryo2Ice estimates) significantly skew the snow depth distributions retrieved. We note that
the number of negative freeboards (20%) is much larger than the 3% negative snow depths reported in Fredensborg Hansen et
al., (2024) which we believe is mostly due to the fact that this study considers a single track averaging averaging over a 300
m and 1-km window  compared to a 7-km window in the aforementioned study. Therefore, we see that the noisy nature of CS2
data especially in landfast ice plays a major factor in the underestimation of the snow depths retrieved from Cryo2Ice.
Differences in the shapes of the distributions from in-situ sites and representative roughness zones of the Cryo2Ice are mostly

a result of the difference in sampling resolutions of Cryo2Ice (~300 m) and the in-situ measurements (5 m). The tails of the
in-situ snow depth distributions (> 40 cm) were largely missed by Cryo2Ice and the Cryo2Ice snow depth retrieval accuracy
is impacted by the presence of sea ice ridges. This impact leads to an artificial widening of the snow depth distributions which
are obtained in the native 300-meter resolution. After adjusting for this difference by averaging both IS2 and CS2 heights over
1-km instead, more realistic snow depth distributions are obtained. We note that while Cryo2Ice generally underestimates
snow depths by 2 to 4 cm compared to in-situ, the 1-km averaged snow depths also show the possibility of overestimation over
significantly rough ice. Therefore, future studies should consider analyzing both the 300 meter resolution product and the 1-
km averaged product in order to get both the meter scale snow depth variations from the 300 meter snow depths as well as the
more representative snow depth distribution from the 1-km averaged snow depths.
Snow geophysical properties, especially snow salinity in the deepest few centimeters of the snowpack, may impact the
dominant scattering surface of the CS2 radar, resulting in the scattering surface shifted upwards into the snowpack, leading
to underestimation of the snow depths. The 1-km averaged       snow depth was slightly underestimated three out of four sites
compared to in-situ measurements; however the median biases compared to in-situ are less than 5 cm. This study identifies
several different sources of uncertainty such as noise in the CS2 heights, surface roughness and snow geophysical properties
which significantly impact the snow depth retrievals in addition to the uncertainty due to the tidal correction. However, it is
difficult to determine given the few centimeters of bias to snow geophysical process, surface roughness and/or errors in the
altimeters' tidal corrections given that a lot of these uncertainties are inter-related and are highly variable among different
length scales. Therefore, a further comprehensive study across different regions is required to isolate the impacts of these
uncertainties and determine their contributions to the total uncertainty. Additionally, there are uncertainties such as the use of
a fixed threshold retracker in CS2 which is not tuned for the landfast sea ice and uncertainties associated with the IS2 fine-
tracker that may also contribute significantly to the snow depth retrievals. Therefore, further studies are required in different
lead-less regions under varying snow conditions for improved insights into the sources of bias in snow depth retrievals from
Cryo2Ice. It is also noteworthy that the suggested method of using ellipsoidal heights from IS2 and CS2 with the tidal
correction may also be applied in regions beyond the landfast sea ice in the Canadian Arctic Archipelago (CAA). However, as
the current method relies on using tidal gauge station data from a nearby station, this method may not be directly applicable
for regions that don't have a tidal gauge station nearby. However, tidal predictions from tide models that consider the impact
of sea ice on the tidal amplitude such as Nucleus for European Modelling of the Ocean (NEMO) may be used instead to
estimate the difference in tides between the passes. While this study suggests the use of Ellipsoidal heights for landfast ice, the
freeboard differencing approach as suggested in Kwok et al., (2020) is better suited for regions where getting a direct estimation
of the sea surface height and direct estimates of the freeboard are available. Findings from this study are encouraging for
estimating snow depth on land-fast sea ice in lead-less regions using Cryo2Ice and for future coincident laser-radar or dual-
frequency altimeter missions.

## Data Availability

ICESat-2 ATL07 data may be accessed from the NSIDC website (See: https://nsidc.org/data/atl07ql/versions/6#anchor-2).

Cryosat-2 data may be accessed from ESA (https://eocat.esa.int/). The snow depth validation dataset is available from the

CanWin Data Hub https://canwin-datahub.ad.umanitoba.ca/data/dataset/cambridge_bay_snowdepth_apr2022.

## Author Contribution

The authors would like to acknowledge Torsten Geldsetzer from the University of Calgary for his input during the planning stages of the Cambridge Bay campaign. We acknowledge Dr. Nathan Kurtz from NASA for providing early ICESat-2 ATL07 release 006 data which was vital for the analysis. Monojit Saha was supported by ArcticNet (Grant Number #52551) and Julienne Stroeve's NSERC Canada 150 Chair (Grant Number #50297). We acknowledge Julienne Strove's NASA Award 80NSSC20K1121, and the University of Manitoba Graduate Student Fellowship (UMGF). We also acknowledge support from ArcticNet Field Aircraft Support for the helicopter support. We acknowledge a Canadian NSERC Discovery Grant to Dustin Isleifson. We also acknowledge John Yackel's NSERC Discovery Grant (RGPIN-2017-04888). Jack Landy was supported by the INTERAAC (airsnow-ice-ocean INTERactions transforming Atlantic Arctic Climate) project under the Research Council of Norway, RCN (#328957), the DynAMIC (Detecting episodes of Arctic sea ice Mass Imbalance) project under RCN (#343069), and the SI/3D (Summer Sea Ice in 3D) project under the European Research Council, ERC (#101077496).

## Competing Interests

At least one of the (co-)authors is a member of the editorial board of The Cryosphere.

## Acknowledgements

The authors would like to acknowledge Torsten Geldsetzer from the University of Calgary for his input during the planning stages of the Cambridge Bay campaign. We acknowledge Dr. Nathan Kurtz from NASA for providing early ICESat-2 ATL07 release 006 data which was vital for the analysis. MS was supported by ArcticNet (Grant Number #52551), Julienne Stroeve's NSERC Canada 150 Chair (Grant Number #50297) and NASA Award 80NSSC20K1121,John Yackel's NSERC Discovery Grant (RGPIN-2017-04888) and University of Manitoba Graduate Student Fellowship (UMGF). We also acknowledge support from ArcticNet Field Aircraft Support for the helicopter support.

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

**Appendix A**
**Table A1: Geophysical corrections applied on the IS2 ATL07 product. The range represents the typical variation in the corrections**
**as reported in the IS2 Algorithm Theoretical Basis Document (ATBD).**

| Geophysical Correction | Typical Range | Source |
|---|---|---|
| Solid Earth Tide | -19 to +27 cm | IERS 2010 (Applied in ATL03) |
| Solid Earth Pole Tides | -0.6 to +0.7 cm | IERS 2010 (Applied on ATL03) |
| Ocean Pole tides | +/- 2 mm | IERS 2010 (Applied in ATL03) |
| Ocean loading | -9.7 to +9.3 cm | GOT4.8 Ocean Tide Model (Applied in ATL07) |
| Ocean Tides | -6.2 to +6.2 m | GOT4.8 Ocean Tide Model (Applied in ATL07) |
| Long period equilibrium tides | -7.1 to +6.0 cm | GOT4.8 Ocean Tide Model (Applied in ATL07) |
| Inverted barometer | -53 to +94 cm | ATL09/GEOS5 FP-IT (Applied in ATL07) |


**Appendix B**


**Table B1: Geophysical Corrections applied in the CS2 Level 2 product. The typical range values are reported in the Cryosat-2 Baseline E Level 2 Product Handbook.**

| Geophysical Correction | Typical Range | Source |
|---|---|---|
| Ocean Tide | -50 to +50 cm | Finite Element Solution FES 2004 Tide Model |
| Long-Period Equilibrium Ocean Tide | < 1cm | Finite Element Solution FES 2004 Tide Model |
| Ocean Loading | -2 to +2 cm | Finite Element Solution FES 2004 Tide Model |
| Solid Earth Tide | -30 to +30 cm | Cartwright Tide model (Cartwright & Edden, 1973) |
| Geocentric Polar Tide | -2 to +2 cm | Historical Pole Positions from CNES |
| Inverved Barometer | -15 to +15 cm | Dynamic Surface Pressure from Meteo France |

**Appendix C**

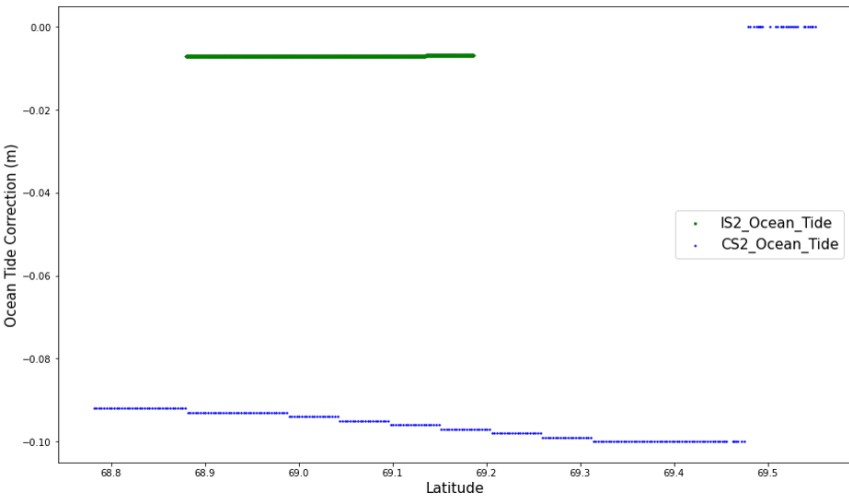


**Figure C1: Ocean tidal correction used in the IS2 and CS2 tracks. The IS2 ocean tide corrections are shown in green while the CS2**
**ocean tide corrections are shown in blue.**
**Appendix D**

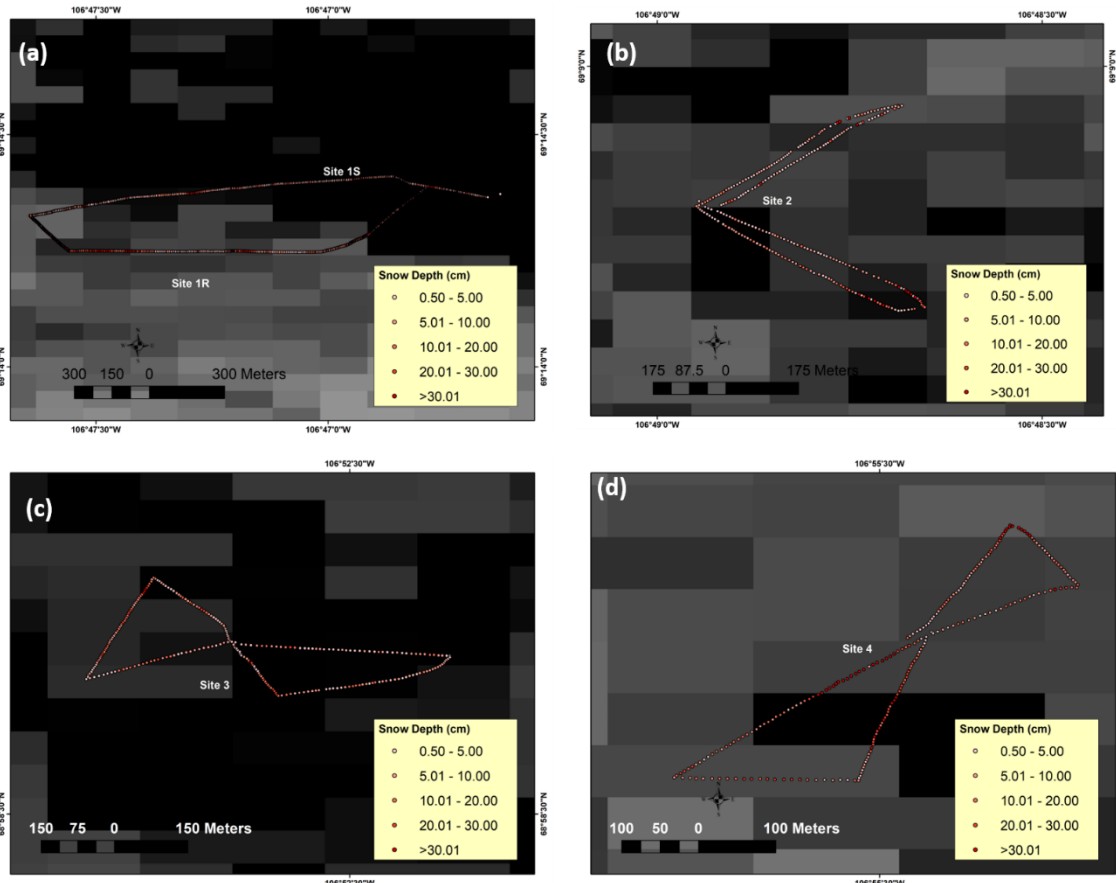

Figure D1: The in-situ snow depth transects conducted in (a) Site 1 (b) Site 2 (c) Site 3 and (d) Site 4. The spatial
distribution of the snow depths are included for each site.
**Appendix E**

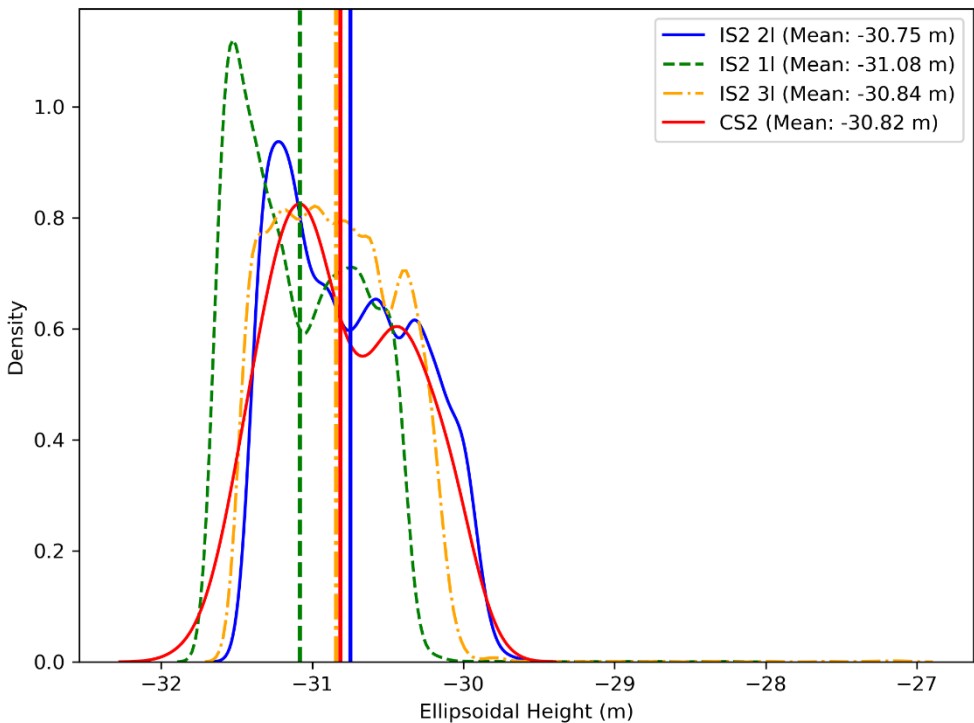


Figure E1: ATL07 ICESat-2 strong beam (IS2 1l, 2l, 3l) sea ice height ellipsoidal height distributions compared to the CS2
height ellipsoidal height distribution.
**Appendix F**
**Table F1 In-situ versus Cryo2Ice snow depth distribution statistics retrieved using 300 meter averaged IS2 and CS2 height**

| | | Mean (cm) | Median (cm) | Lower Quartile (cm) | Upper Quartile (cm) | Inter-quartile range (cm) |
|---|---|---|---|---|---|---|
| Site 1 | In-Situ | 12.2 | 7.8 | 4.1 | 16.3 | 12.2 |
| | Cryo2Ice | 4.7 | 4.9 | -1.8 | 9.8 | 11.6 |
| Site 2 | In-Situ | 9.7 | 5.2 | 3.7 | 9.2 | 5.5 |
| | Cryo2Ice | 1.9 | 4.8 | -5.9 | 8.5 | 14.4 |

| | | Mean (cm) | Median (cm) | Lower Quartile (cm) | Upper Quartile (cm) | Inter-quartile range (cm) |
|---|---|---|---|---|---|---|
| Site 3 | In-Situ | 8.9 | 6.9 | 4.2 | 11.9 | 7.7 |
| | Cryo2Ice | 0.61 | 3.4 | -5.4 | 5.8 | 11.2 |
| Site 4 | In-Situ | 17.1 | 13.8 | 6.7 | 22.4 | 15.7 |
| | Cryo2Ice | 10.6 | 8.3 | -0.6 | 18.5 | 19.1 |



**Table F2 In-situ versus Cryo2Ice snow depth distribution statistics retrieved using 1-km averaged IS2 and CS2 height**

| | | Mean (cm) | Median (cm) | Lower Quartile (cm) | Upper Quartile (cm) | Inter-quartile range (cm) |
|---|---|---|---|---|---|---|
| Site 1 | In-Situ | 12.2 | 7.8 | 4.1 | 16.3 | 12.2 |
| | Cryo2Ice | 7.1 | 6.3 | 4.6 | 8.8 | 4.2 |
| Site 2 | In-Situ | 9.7 | 5.2 | 3.7 | 9.2 | 5.5 |
| | Cryo2Ice | 4.0 | 4.9 | -8.4 | 8.2 | 16.6 |
| Site 3 | In-Situ | 8.9 | 6.9 | 4.2 | 11.9 | 7.7 |
| | Cryo2Ice | 6.5 | 2.3 | -1.7 | 3.8 | 5.5 |
| Site 4 | In-Situ | 17.1 | 13.8 | 6.7 | 22.4 | 15.7 |
| | Cryo2Ice | 18.7 | 8.3 | 15.1 | 24.2 | 9.1 |
