# Peer review of "satellite observations"

_EGUsphere, 2023_

## Author Response (AR1)

**Response to the Editor**

Dear author,

Both referees have major concerns and suggested accepting the study only after major revisions and have provided critical comments and questions. You have responded to the comments and questions but in my opinion you have not always sufficiently addressed all raised issues, e.g. about the retracker.
Personally I am most concerned about the fact that different geophysical corrections have been applied for the IS2 and CS2 data. Together with the CHS tidal correction it is difficult for me to understand where the largest uncertainty comes from. Another question is the comparison with the C-band SAR backscatter which I suppose has only limited meaning given the large spread of the distribution representing a large range of different sea ice and snow conditions.

Best regards

Lars Kaleschke

*The authors thank the Editor for their valuable comments and queries. As part of the re-submission, we have tried to address all major issues raised by the reviewers. Please note that major adjustments including the consideration of the negative snow depths as well as 1-km resampled Cryo2ice have been added based on feedback from both reviewers. We believe the error distribution is better reflected with the new adjustments giving a complete picture of the range of uncertainties. However, it is difficult to ascertain the exact contribution of these uncertainties given a significant number of these uncertainties are related to each other as well as varies over scale and time.*

*With a fixed threshold retracker we cannot be sure that it is tuned to match the local ice conditions, for instance the typical surface roughness of landfast sea ice. It would be valuable to compare the performance of different retrackers, including empirical fixed threshold versus physical retrackers, over landfast sea ice surfaces. However, given the scope of the study is to validate the snow depth products retrieved from Cryo2Ice, we believe this needs to be considered as a separate study. We decided to go for the fixed threshold retracker (70%) which is most widely used in the CS2 Baseline E product since this is the recommended retracker for the SARIn mode over sea ice. The CCI+ Climate Record Data Product (CRDP) is a 50% fixed threshold retracked CS2 product which is only available between 2020-2021 and therefore not available for the time period of this study (April 2022).*

*Please note that the geophysical corrections need to be different between the sensors, like the atmospheric attenuation / ionosphere correction for CS2. The time difference between*

*the two passes also means that other corrections, like the tidal and IB corrections, which have a time dependency, could be different for each sensor. So, it wouldn't make sense to apply e.g. all the CS2 corrections to IS2, or vice versa. Therefore, we opted to keep the same tidal corrections for both IS2 and CS2 as provided in the products but opted to find the bias in tidal correction between the sensors by comparing to the CHS data. Here, the assumption is that the difference in tidal corrections between the IS2 and CS2 passes should reflect the difference in water level that has occurred between the passes i.e. within the 77 minutes between the IS2 and CS2 pass. This difference in water level is then retrieved from the CHS data and compared to the difference in tidal correction applied in IS2 and CS2.*

**Response to Reviewer 1** "Snow Depth Estimation on Lead-less Landfast ice using Cryo2Ice satellite observations" by Saha et al.

Summary:

The study assesses the potential for near-coincident ICESat-2 and Cryosat-2 (Cryo2Ice) satellite data in estimating snow depth over landfast ice in the Canadian Arctic Archipelago. Snow depth is retrieved by calculating the absolute difference in surface height from the two satellites, considering an ocean tide correction. The study compares the retrieved snow depths from Cryo2Ice with in-situ measurements, showing good agreement in terms of mean values. However, Cryo2Ice snow depths were, on average, underestimated by 20.7%. Discrepancies are attributed to differences in sampling resolutions, snow characteristics, surface roughness, and tidal correction errors. The results suggest the potential for estimating snow depth over lead-less landfast sea ice, but further investigation is needed to understand biases related to sampling resolution, snow salinity, density, surface roughness, and altimeter correction errors.

*The authors thank the reviewer for the very comprehensive and accurate overview of the study.*

General Comments:

This is an interesting study, which will be valuable to improve our understanding with respect to retrieving snow depth from a dual-altimeter approach. I had no problems to follow the paper, but I believe clarity can be improved. However, there are some parts in the analysis and discussion, which I think need some clarification and revision. I think this work deserves publication, but major revisions are needed.

My main concerns are:

- There is quite some focus on the tidal correction, which I agree is important. But one of the main limitations, from my point of view, is the **large CS2 footprint and the noise in the CS2 height retrievals**, which is not surprising as we know from previous studies. But considering the relatively small sample size, it will have a large impact. The reasons for the CS2 height uncertainties are discussed by the authors, e.g., the surface roughness, snow salinity, scattering horizons in the snow layer. And given the significant difference in

footprint between IS2 and CS2, we can hardly assume that CS2 heights will represent the corresponding snow-ice interface, even if colocated. Considering the snow depth distribution from in-situ measurement sites indicates how much spread is just within one CS2 footprint. In addition, the retracker used for the CS2 height retrievals is a threshold retracker, using a fixed threshold, where we cannot be sure if it is tuned exactly for the same ice conditions we find in this area. I believe this study can contribute to characterize and **quantify uncertainties and limitations of this approach**, but it **should be emphasized more clearly**. For example, I think the comparison between the mean/median values of Cryo2Ice and in-situ measurements has only limited meaning, which brings me to my next point.

*The authors thank the reviewer for their valuable observations on the analysis. The authors agree that the difference in CS2 and IS2 footprint is definitely one of the major challenges of getting coincident snow depths from Cryo2Ice. Studies often deal with this by smoothing the CS2 data. A recent study by Fosberg et al. (2024) chose to smoothen the CS2 with a 3500m radius (~7 km moving average window) to remove speckle and other random sources of noise. However, given the fact this study considers snow depth over ~75 kilometers and tries to identify the uncertainties in combining CS2 and IS2 over landfast sea ice, a smoothening approach may lead to loss of distinctive roughness features and also a reduction in overall resolution of the snow depth retrieved. Averaging IS2 over ~300 metre along-track resolution allows us to make the closest comparison to the 300 meter along-track footprint of CS2 and ~200 meter in-situ snow depth retrievals. However, the current revised paper includes a 1-km averaged CS2 and IS2 snow depths included as an adjustment to account for the mismatch in the footprint size. We notice that smoothing the ATL07 data over ~1km makes us loose some of the distributive ridges which IS2 is very sensitive to (Figure R1). We note that averaging the CS2 and IS2 data over 1-km does indeed make the snow depth distributions more realistic as presented in Section 4.3 The authors agree with the reviewer that further emphasis on the impact of the difference in footprint and lines 422 to 428 addresses this.*

*The authors do agree that some of the statistical interpretations made based on the mean snow depth retrievals need to be revised. Therefore, the current analysis focuses more on the median bias and comparing the distributions of the Cryo2Ice and in-situ rather than focusing solely on the mean bias which is severely impacted by the high negative snow depth values.*

[Figure]

Figure R1: IS2 and CS2 Height distributions. IS2 2l height in native distribution is shown in green while CS2 height is shown in purple. The blue line shows the smoothed IS2 and CS2 lines over 1-km.

- I am not an expert in statistics, but some of the decisions made in the processing need some clarification and potential revision. I do not think it is a good idea to **just drop negative snow depth values.** From a physical point of view, they do not make sense, but statistically they are important, **reflecting the impact of uncertainties**. By removing only negative values, your snow depth retrieval is likely biased towards higher values. The negative values should be **also part of all the histograms** because that's the reality when you subtract one height retrieval from another, when both come with uncertainties. Moreover, some steps and figures need to be explained in more detail, see therefore the specific comments below.

  *The authors agree with the reviewer that it is important to include the negative snow depth values as part of the final computation of the snow depths and therefore has been considered as part of the revised snow depth estimations. However, the impact of large negative snow depths caused largely due to the noise of CS2 needs to be discarded in the final snow depth estimation. As shown in Figure R2, there is a significant portion of negative snow depths higher than 2 standard deviations from the mean which also corresponds with noisy portions of the CS2 ellipsoid height which ultimately biases low the mean snow depth retrievals as shown in Figure 8. Therefore, we disregard negative snow depths that are lower than 2 standard deviations from the mean and attribute this to the uncertainty due to the CS2 noise. After adjusting for the outliers in the negative snow*

*depths, we get a mean snow depth distribution of 7.44 cm. The new distribution of the Cryo2Ice snow depths are presented in Figure 9.*

[Figure]

*Figure R2: Distribution of raw Cryo2Ice snow depth values including negative snow depth values*

Specific Comments:

L101: Figure 3 is introduced before Figure 2.

*This was a mistake on part of the author. The Figure number has been changed to Figure 1 instead of Figure 3.*

L113: For the snow depth measurements, what was your sampling strategy? Can you be more specific here? Did you walk straight transects? Did you ensure representative sampling, considering the fraction of deformed sea ice?

*The authors thank the reviewers for their question regarding the sampling strategy. The authors agree that further details about the sampling strategy are warranted in the paper. The transects were set considering wind direction as well as the sea ice surface features for each spot. As shown in Figure R4 (which will be included in the appendix of the revised paper), the shape of the transects are demonstrated along with the representative snow*

*depths for each site. The sampling strategy was to ensure that we cover Cryo2Ice along-track and across-track directions, wind direction and sample the different representative roughness features. In Site 1, two L-shaped transects representing the rough and smooth sea ice zones were conducted (Figure R4 (a). For Site 2, which had a large ridge which covered a significant portion, we tried to take two different L-shaped transects crossing the ridge at least four times but ensuring we get the near-ridge features as well as the smoother zones further away from the ridges (Figure R4 (b)). For Site 3 and 4 which had wider regions of smooth and rough sea ice respectively, two L-shaped transects were conducted (Figure R4 (c) & (d)).*

[Figure]

*Figure R4: The in-situ sampling sites showing the position of the magnaprobe samples for (a) Site 1R and 1S, (b) Site 2, (c) Sites 3 and (d) Site 4.*

L157: The MSS is not mentioned under 2.6. I suggest to briefly explain the reason here.

*The line now reads: "However, the mean sea surface (MSS) ensuring that the IS2 ATL07 heights are referenced to the WGS84 ellipsoid. The MSS is calculated based on decadal averages and therefore are not representative of the variation of sea surface heights within the 77 minutes' interval between the IS2 and CS2 passes".*

L164: To my knowledge, the ATL07 product does not contain the individual photon heights, but segments of different length that aggregate the photon heights from the ATL03 product. I assume you have used these segments?

*Yes, we have used the aggregated photon heights used in the ATL07 product. On average each segment i.e. difference between two aggregated ATL07 heights is 8.3 meters. We have made this clear using the following revision in L71 "**ATL07 Heights are aggregated from ATL03 photon heights over variable distances, the heights were aggregated over 8.3 meters on average over the portion of the track used in this study**."*

L167: Are the retrieved Cryo2Ice snow depths not arranged along a straight line? Why then investigating spatial autocorrelation? Isn't it nearly 1D? Moreover, when I look at Fig. 8 (bottom plot), I find it hard to imagine how this works. The sample size is not very high and there is a lot of noise. And the spacing between point is already 300 m. Can you show a variogram? (Just in the response, does not need to go into the manuscript).

*The authors agree that only the along-track variability in snow depth is being represented in the variogram given that the snow depths are collocated to the CS2 POCA points. However, the variogram gives us a sense of the spatial autocorrelation along-track which is useful to decide on which lag distance the snow depth points are auto correlated. Therefore, we observe that the variogram in Figure R4 starts to reach an inflection point around 300 meter and reaches a sill ~ 1km. We also notice that beyond the 1-km lag distance, the semi-variance starts to become constant. Due to the high variability in the data, the variogram analysis doesn't provide a concrete lag distance but does give an indication that beyond the 1 km point, the spatial autocorrelation would be negligible. Therefore, while we don't believe the variogram gives a concrete evidence of which lag distance to consider for averaging, we can say that beyond the 1km distance, the snow depth points are not autocorrelated.*

[Figure]

L196: That's a nice approach with the Sentinel-1 backscatter. I suggest checking the "stability" and representativeness of the ICESat-2 heights, making use of the other beams. Just compare the height distributions from the 3 strong beams for the area of interest.

*The authors thank the reviewer for their valuable suggestions. We compared the IS2 height distributions which are presented in the Figure R6. It is noteworthy that IS2 2l strong beam's height distribution is most similar to CS2 height distributions. This is because the distance of the strong beam is within ~1.5 kilometer of the CS2 Points of Closest Approach (POCA). Therefore, we believe using only the IS2 2l strong beam which is the closest to the POCA CS2 would make the best colocation of the IS2 and CS2 tracks. This is based on the assumption that the snow distribution corresponding to the CS2 POCA and the closest IS2 strong beam would be most similar. This assumption is further tested by comparing the Sentinel-1 Backscatter retrieved from all the IS2 strong beams and CS2 (Figure R7). We notice that while the mean ellipsoidal heights are similar, there is significant differences among the distributions of both IS2 1l and IS2 3l strong beams. Therefore, we don't consider the IS2 1l and IS2 3l strong beams for the subsequent colocation and snow depth retrieval steps.*

[Figure]

*Figure R6: IS2 strong beam height distributions*

[Figure]

*Figure R7 Comparing the backscatter retrieved from all IS2 strong beams with CS2*

Figure 4: I suggest changing the legend. It is misleading. It looks like the backscatter of IS2/CS2 is shown here…

*Noted. We changed the legend and the caption will also elaborate on this.*

L206: Is this related to Figure 11? May be show this together with Figure 4? Farrell et al. (2020) primarily use ATL03. The ATL07 segments can be quite long. How many segments do you get on average within the 300 m segments? Can you derive a meaningful roughness from this?

*L206 was intended to introduce surface roughness as a potential uncertainty impacting retrieved surface roughness. In order to compute surface roughness, Farrell et al., (2020) uses ATL07 segments but over 25 km long segments. Given our study area is ~75km long, this approach would not be able to different the difference in roughness zones. Therefore, we compute the roughness over ~300 metres instead. We get 36 ATL07 height segments within each 300 metre segments for the portion of the IS2 track that was studied. Therefore, each ATL07 segment is ~8.3-metre-long which is sufficient to portray the regional (~200 meters for each site) difference between each surface. The University of Maryland- Ridge Detection Algorithm (UMD-RDA) in Farrell et al., (2020) uses the ATL03 photon heights*

*retains much finer resolution (0.7 metre) which is aimed towards ridge detection. However, we are more interested in representative roughness over ~200 metre regional sea ice roughness zones (Sites 1 to 4) which we believe is sufficiently captured by the 8.3 metre resolution of the ATL07.*

L236: Figure 7 -> Figure 6?

*The authors thank the reviewer for pointing out the incorrect Figure number. Figure number changed to Figure 6 instead of Figure 7.*

Figure 5: The blue line is not explained.

*The blue line is the Probability Density curve for the respective distributions. We will elaborate this in the caption.*

L251: I don't see the negative values in Figure 8. I suggest adding a class with a specific colour for values <0. From Fig. 7, it does not look like negative values primarily occur close to the coasts.

The authors agree with the reviewer that the negative values need to be included in the figure and Figure 7 has been revised accordingly. The authors agree that the distribution of the negative snow depths are more random and we do notice that there are greater concentrations closer to the coast, but we decided to exclude making a generalization.

L252: I would argue that with removing negative values, you introduce a positive bias in the snow depth retrieval. It will only make sense if you assume that underlying uncertainties affect the snow depth exclusively in one direction. But looking at Figure 7, it just seems that there is significant noise on the CS2 heights, which goes in both directions (positive and negative).

The authors agree with the reviewer that the negative values need to be included in the subsequent snow depth calculations and the uncertainty calculations. Including the negative snow depths (within 2 standard deviations of the mean) has significantly changed the error estimates which have been presented as part of the revised paper.

L295: I haven't fully understood why this test is done. "The test results show significant difference between in-situ sites which was also evident in the corresponding Cryo2Ice snow depths." Which are the corresponding Cryo2Ice snow depths? I guess there are just a handful in the vicinity of each site?

*The Kruskal-Wallis non-parametric test was conducted to test whether the retrieved Cryo2Ice snow depths have similar site to site differences as obtained from the field. The test answers the question 'Site 2 has significantly different snow depths compared to Site 1 as seen from the field data, is this also true for the Cryo2Ice snow depths from Site 1 and 2?'. The test is important to check if the retrieved Cryo2Ice snow depths are realistic. The corresponding snow depths are the closest Cryo2Ice snow depth points which are within a similar roughness zone (Figure 8- Yellow lines). The roughness zones are defined for each site as the length of the Cryo2Ice track that has roughness (calculated from IS2) within 1*

*standard deviation of that obtained from the Site. Each Site has different number of Cryo2Ice snow depths that are within the same roughness zone.*

L299: Related to the previous question: How many Cryo2Ice snow depths are you using for the comparison?

*The number of Cyo2Ice snow depths varied based on the definition of the similar roughness zones (Figure 8, Yellow Lines). That is, the Cryo2Ice snow depth points which fall within 1 standard roughness are considered corresponding to each site. The number of Cryo2Ice snow depth sites compared to each site is presented in the Table below:*

*Table R1: The number of Cryo2Ice snow depths are within the similar roughness zones for each site*

| Site | Number of corresponding Cryo2Ice snow depths |
|------|----------------------------------------------|
| Site 1 | 11 |
| Site 2 | 19 |
| Site 3 | 30 |
| Site 4 | 7 |

Figure 10: I suggest showing the "raw" distributions, not the density functions. Again, how many Cryo2Ice samples have been used at each site for the PDFs?

*Given the difference in frequency between the number of In-situ sites and the Cryo2Ice, the visualization of the raw distributions is challenging and therefore the density functions are used. The number of Cryo2Ice samples will be included in the PDFs. Table R1 provides the number of Cryo2Ice snow depths used for each site.*

Figure 7: It would be also interesting to see the IS2 heights from the co-registration, averaged on the 300 m segments. Perhaps you can add them here?

*The IS2 300 meter segment heights have been included in Figure 7.*

Figure 8: I suggest adding the mean and standard variation from the in-situ measurements at the 4 sites.

*Noted. The mean and stand variations will be added to the to Figure 8.*

L363: $R^{**}2 = 0.04$ basically means no correlation I believe. But considering the noise level, especially from the CS2 heights, and the relatively low sample size, I wouldn't expect a higher R here.
*The authors agree with the reviewer and the line is revised to:*
*"There was no significant correlation ($R^{**}2 = 0.04$) between IS2 and Cryo2Ice snow depths"*

**Reviewer 2**

This study presents a first estimation of snow depth from near-coincident spaceborne dual-frequency (laser and radar) altimetry over lead-less landfast ice in the Canadian Arctic Archipelago. The authors utilize one CRYO2ICE track (separated by 77 minutes and approximately 1.5 km), and estimates snow depth from ellipsoidal elevations after applying a tidal correction to the elevations. This tidal correction allows for derivation of snow depth from the difference in elevations assuming laser (ICESat-2/IS2) is reflected at the top of the snowpack/air-snow interface, and radar (CryoSat-2/CS2) is reflected at the snow-ice interface over lead-less sea ice. However due to the non-presence of leads, the impact of tides is accounted for through comparing the changes to water level caused by tides using the models from the satellite data with tide gauge estimates. The derived CRYO2ICE snow depth estimates (with negative snow depths removed) are compared with a dedicated ground-based campaign along the orbit.

An interesting paper that provides some interesting results to the dual-freq. snow depth approach – and, with the limited reference data available to compare with, especially along CRYO2ICE orbits, this paper can provide some interesting insights along this one orbit. The inclusion of much needed in situ reference data along such an orbit feed into some good discussion topics and the paper warrants publication. The paper read easily, although I have a few comments regarding the figures and tables, which I hope the authors will consider. Furthermore, I still have a few major points to be addressed before publication, as I believe there are more work required to ensure that CS2 and IS2 are comparable at the resolution used in the study, that the tide correction is applicable beyond just this example (and if not, that this is further discussed), and whether some of the assumptions of this study holds up.

*The authors thank the reviewer for their comprehensive and insightful overview of the paper and appreciation of the topic.*

**Major comments**

*Tide correction and discussion relating this to different ice regimes/areas*

I've yet to encounter this methodology before, but I am intrigued by it, and wonder how this may be applicable on a larger scale. However, I am not fully convinced by the correction applied as of now, was somewhat confused by it, and hope the authors could provide some more information regarding the ocean tides used in the study. You state that there is an average difference along track of 7.9 cm (**on the co-registered points or? – and if so, how are the IS2 ocean tide of co-registered points even computed?**) but the difference in water level from CHS was 6 cm. This is only compared to one point (tide gauge) vs. full along-track data. Could you provide a figure (maybe just in response to reviews, but potentially also in the manuscript) of the difference in ocean tide models along-track (maybe not using the average value, but maybe maximum and minimum too/or show distributions). This would also feed into your statement about the 1.9 cm correction (when compared with CHS water level) representing the systematic bias – this seems somewhat low, when we see that the ocean models may differ by +/- 12 cm (ranges +/- 50 cm vs +/- 62 cm) along the track. I think the study would greatly benefit from some more results and discussions on this.

*The authors thank the reviewer for finding the methodology interesting and agree that some further details and analysis need to be presented to expand on the methodology accounting for the difference in ocean tides between the IS2 and CS2 passes. The basic assumption of this methodology is that the difference in the average ocean tide correction applied in IS2 and CS2 respectively account for the difference in water level recording from the tidal gauge dataset in the same period.*

*We plan to replace Figure C1 with the following figure comparing the along-track IS2 and Cs2 ocean tide:*

[Figure]

*Figure R1 Along-track Ocean Tide Correction applied in the(a) IS2 and (b) CS2 products*

*The tidal corrections for collocated IS2 and CS2 are obtained from the tidal corrections provided for each product. For ATL07, a tidal correction value is provided for each segment which is then averaged over 300 meter. Similarly, the ocean tide correction applied in the CS2 Level-2 product is also provided in the 1-Hz Cryosat-2 Level 2 product which we converted to the 20 Hz level in order make this comparable to the 20Hz height product. Therefore, after collocating the two tracks based on latitude, we compare the 300 meter averaged tide correction IS2 with the collocated CS2 ocean tide corrections.*

*The maximum along-track IS2 ocean correction applied was 0.7250 cm- 0.700 cm whereas the CS2 ocean correction applied varied between 9.20 and 10.00 cm. Therefore, the ocean tide correction doesn't vary significantly along-track i.e. less than 1 cm for both IS2 and CS2. Also, we currently don't have tide data that may account for the less than 1 cm difference along-track, considering the average along-track relative tidal height difference between the 77 minutes of the passes seems reasonable in the context of landfast sea ice. Please note that the current technique relies on the assumption that landfast sea ice undergoes insignificant drifting and the difference in vertical height is accounted for in the relative difference between the tidal corrections in IS2 and CS2. Therefore, we assume that the relative change in water level will vary insignificantly along ~75 km track for the 77 minutes between the passes. We consider the average difference in the difference in tidal correction (IS2-CS2) and compare it to the difference in water level from the tidal gauge station.*

*Subsequently, we compute the along-track height difference between collocated IS2 and CS2 has an average value of 7.9 cm which is then compared to the 6.0 cm tidal gage difference. Please note that this method doesn't rely on the absolute tidal corrections applied which can vary widely globally. The difference between the IS2 and CS2 tidal correction range (i.e.+-62 cm and +-50 cm) are the ranges over which the global tidal models vary and since we are not correcting any individual model, our correction which is computed specifically for this particular track is not comparable. Please note that if this method was applied in a region where the tidal correction values were higher, the bias could potentially be higher compared to the closest tidal gauge data.*

In addition, I appreciate that the study investigates only the snow cover on landfast ice along the one track where you acquired ground observations. However, this methodology of applying a tide-correction is interesting for coastal and landfast sea ice beyond just the Canadian Archipelago. The manuscript would benefit from providing a greater perspective on applying this on a larger scale (across the Arctic, discussing whether ellipsoidal height differencing is better/equal to freeboard differencing etc.). If possible, would be interesting to see if it was possible in e.g., the Laptev Sea or similar – and also, in areas with no tide-gauges to compare with. Or at least a discussion about this would be very interesting.

*The authors appreciate that the reviewer's find this an interesting method to test in other geographical conditions with varying conditions. The authors agree that this method warrants further investigation perhaps in different parts of the Arctic such as the Laptav Sea. However, the authors think that a comprehensive validation campaign in such conditions similar to the one presented in the paper is needed for making concrete statements. Therefore, such campaigns may be proposed as part of future studies.*

*The conclusion section Line 450 to 457 include discussion about the applicability of this method in zones which don't have tidal gauge stations.*

Similarly, I believe that this work on landfast ice is important, but would have expected a discussion relating it to the bigger picture and sea ice in general (landfast and drifting) – **could you provide some insights/relate it to e.g., deriving this in drifting sea ice in the Arctic/Antarctic?** I acknowledge the very different sea ice/atmospheric patterns here, but I believe that this warrants some more discussion or at least, additional acknowledgements of the limitations/difficulties/differences when deriving this over landfast ice vs. drifting sea ice, beyond just the limitations regarding leads only.

*The author's appreciate the reviewer's suggestion to include a discussion about the bigger picture i.e. landfast vs drifting sea ice in the Arctic/Antarctic. However, one of the critical assumptions made in this paper was that landfast ice in the narrow channels of the Canadian Arctic Archipelago during a significant portion of the winter months don't have significant drifting and therefore the small temporal difference between the IS2 and CS2 may be assumed to be equivalent to the variation in ocean tide. In case of the drifting sea ice in the central Arctic, we believe various other factors including ocean currents and other dynamic sea ice processes will cause the sea surface height to change significantly even between the IS2 and CS2 passes. Therefore, we will need to have direct altimeter derived estimates of the sea surface height to be able to compute freeboards in case of the drifting sea ice in the Central Arctic.*

*Spatial scales, C2I snow depths and smoothing IS2 elevations*

I believe there is more work to be done to ensure that differences in footprint and resolutions have been thoroughly considered. The fact that we are observing at much different scales (300 m x 1600 m vs. ATL07's varying resolution), simply smoothing IS2 to 300 m does not seem sufficient. Also considering the average distance of IS2 and CS2 being 1.5 km, I believe there needs to be done more work on the IS2 data to ensure that they have processed to be "comparable". **I suggest having a look at the data from the other strong beams, to investigate the variability along the IS2 tracks and from here, you may actually also see if they are seeing "similar surfaces/distributions" which you've otherwise support using the Sentinel-1 data. In addition, providing some statistics on segment lengths from IS2 would be great to fully understand the coverage of IS2 when smoothing and co-registering to CS2**.

*The authors thank the reviewer for their valuable suggestions. We compared the IS2 height distributions which are presented in the Figure R2. It is noteworthy that IS2 2l strong bema's height distribution is most similar to CS2 height distributions. This is because the distance of the strong beam is within ~1.5 kilometer of the CS2 Points of Closest Approach (POCA). Therefore, we believe using only the IS2 2l strong beam which is the closest to the POCA CS2 would make the best colocation of the IS2 and CS2 tracks. This is based on the assumption that the snow distribution corresponding to the CS2 POCA and the closest IS2 strong beam would be most similar. This assumption is further tested by comparing the Sentinel-1 Backscatter retrieved from all the IS2 strong beams and CS2 (Figure R3). We notice that while the mean ellipsoidal heights are similar, there is significant differences among the distributions of both IS2 1l and IS2 3l strong beams. Therefore, we don't*

*consider the IS2 1l and IS2 3l strong beams for the subsequent colocation and snow depth retrieval steps.*

*Further, the Cryo2Ice snow depths are now adjusted for the difference in footprint size by considering 1-km averaged IS2 and CS2 heights for calculating snow depths (Section 4.3). This significantly improves the Cryo2Ice distributions making them more realistic compared to in-situ.*

[Figure]

*Figure R2: IS2 strong beam height distributions*

[Figure]

*Figure R3 Comparing the backscatter retrieved from all IS2 strong beams with CS2*

*The IS2 300 meter length segments have been added to Figure 7.*

In addition, **I would have liked to see the semi-variogram (or just variogram)** that you have created showing this spatial autocorrelation of 1 km of the in-situ snow depth distribution. That somewhat counters the assumption that IS2 and CS2 are seeing similar snow – or at least similar snow variability (as they are separated by 1.5 km). The assumption that smoothing IS2 to CS2's along-track resolution of 300 m is sufficient is not well supported, **as studies on along-track radar altimetry data usually apply some level of smoothing to the along-track radar data due to the noise (which is evident in your figure too).** The CS2 data is – to a large extent – impacted by noise, off-nadir reflections (of ridges, leads and other) etc., which you see in Figure 7. This might also explain your "lack" of correlation with IS2 roughness. Ideally, we would expect less difference in the CS2 ellipsoidal heights along the orbit than in the laser if there is full penetration to the snow-ice interface, but that is not the case. A value of $R2=0.04$ basically states that there is no correlation. However, I also do not think – based on the ellipsoidal heights already shown – that we would expect this due to the noisy behavior of CS2. Another comparison you could use to identify what is primarily contributing to the variability in your snow depth estimates, is **computing the correlation between IS2 ellipsoidal heights/C2I snow depth and CS2**

**ellipsoidal heights/C2I snow depth (or another measure of the CS2/IS2 elevations that are of smaller magnitudes than ellipsoidal heights along your track)**.

I do believe there is room for more speculation/discussion related to the C2I snow depths and how they compare with the in-situ data – just comparing mean/median seems … insufficient or at least, as if there might be more to discover, especially since the distributions are different even if the average values are similar. **But I'll leave that up to you to see if you think there is more to extract from these plots and/or analysis**.

*The authors thank the reviewer for their valuable comments and suggestions. The authors realise that only the along-track variability in snow depth is being represented in the variogram given that the snow depths are colocated to the CS2 POCA points. However, the variogram gives us a sense of the spatial autocorrelation along-track which helps in understanding the overall variability. Therefore, we observe that the vairogram in Figure R4 starts to reach a inflection point and reaches a sill ~ 1km. We also notice that beyond the 300 meter lag distance, the semi-variance starts to become constant. Due to the high variability in the data, the variogram analysis doesn't provide a concrete lag distance but does give an indication that beyond the 1 km point, the spatial autocorrelation would be negligible.*

[Figure]

*Figure R4 Semi-variogram analysis of the Cryo2Ice snow depths. Y axis represents the semi-variance while the x-axis shows the lag-distance.*

*In order to test the impact of smoothening the CS2 footprints to reduce noise, a 1-km filter was applied to both IS2 and Cs2 native heights (Figure R5). We notice that averaging over 1-km causes IS2 to be less sensitive to the roughness features which is very well represented in the native resolution. We also notice that some of the noise if still present*

*after applying the smoothening. Therefore, the approach that we take is to consider the finest possible resolution i.e. 300 meter corresponding to the CS2 along-track footprint size and then to include the impact of noise which is reflected as an uncertainty in the revised analysis. This allows us to understand the impact of major differences in roughness which characterize each in-situ site surveyed. Therefore, adopting the finest possible resolution allows us to understand the impact of roughness features on the final snow depth outputs while also accounting for the uncertainty due to noise.*

[Figure]

*Figure R5 IS2 and CS2 Height distributions. IS2 2I height in native distribution is shown in green while CS2 height is shown in purple. The blue line shows the smoothed IS2 and CS2 lines over 1-km.*

*The authors thank the reviewer for the suggestion for computing the correlation between the snow depths retrieved and individual ellipsoidal heights. We see that the Adjusted C2I snow depths only have weak correlation to IS2 and CS2 ellipsoidal heights. We note that CS2 heights have a more significant correlation coefficient to the retrieved snow depths and therefore have a much larger impact on the retrieved snow depths compared to the IS2 heights.*

| | Correlation Coefficient |
|---|---|
| IS2 Ellipsoidal Height vs Cryo2Ice Adjusted Snow Depths | -0.1213 |
| CS2 Ellipsoidal Height vs Cryo2Ice Adjusted Snow Depths | 0.2509 |

*The correlation between IS2 and CS2 ellipsoidal height has been included in the statistical analysis Lines 281 to 283.*

**Minor comments**

Figures:

- Tables in figures should be removed (following TC policy) – either make the tables into individual tables, or simply include the information in the legend/on the figure (since it is primarily average values and such).

  *The authors thank the reviewer for pointing this out. The figures and tables will be adjusted to be compliant with TC guidelines.*

- In addition, have a look at the resolution of the figures (e.g., Figure 11) and consider improving the DPI for better visualization.

  *The resolution of the new figures will be adjusted.*

- Consider using the same color for the line representing locations of different sites across figures – it will make comparison between figures (e.g., Figure 7, 8 and 11) easier. Also, double-check that your figures work for readers with color deficiencies.

  *Noted. The color consistency will be considered while revising the figures. The figures are color blind friendly.*

  Tables:

- In general, there are quite a few tables that provide limited insights which could just as easily have been mentioned in the figures and would likely provide some more connection having it next to e.g., the distribution figures. I encourage you to reconsider the number of tables you have, and whether the content of the tables could be put in the figures instead.

  *The authors thank the reviewers for the suggestion to include some of the numbers along with the distribution graphs. The number of tables has been reduced.*

  Data policy/statement:

  "Available upon request" does not follow TC guidelines. Please provide a DOI for where to obtain the data – either as raw data, or as the processed data that you are presented in the manuscript. In addition, I am not able to open the link for the CryoSat-2 data – in addition, I have not encountered this website before (you did not use the science server or FTP site?). Please have a look at this link again to ensure that the link is active.

  *The authors thank the reviewer for pointing out the inconsistency with the TC data distribution policy. The raw data will be made available and shared in the revised manuscript. The link for the Cryosat-2 data seems to be active, however, the reviewer may*

*try to access the data using the following link: https://eocat.esa.int/sec/#data-services-area. The EO-CAT server was used to download the Cryosat-2 data.*

**Specific comments**

Line 71-73. Do you include low-confidence data (that is, ATL03 flag of low confidence or another ATL07 flag) across the entire track, or only close to the coast – or what is meant by low confidence? Could you provide a measure of the amount of low-confidence data or was it flagged somehow? And where this low confidence is along the track?

*ATL07 corrected heights are not provided within 25 km distance from the coast primarily due to the low-confidence in the tidal models close to the coast. However, The ATL07 Version 6 product includes an uncorrected product that includes the low-confidence heights within the 25 km buffer from the coast as well. The portion of the track with low-confidence data is demonstrated in the Figure 1 below:*

[Figure]

*Figure R6 Spatial Distribution of the Corrected and Uncorrected IS2 data used.*

*Please note that the uncorrected IS2 data shown in Figure refers to the raw ATL07 uncorrected data which is available in ATL07 Version 6. Therefore, approximately 37 Kilometers of the track close to the land fall under the low-confidence of tidal model zones*

*close to the coast. Therefore, due to this uncertainty, it is important to correct the tidal corrections and ensure that the relative difference between the tidal corrections applied in IS2 and CS2 is similar to the change in water level from tidal gauge stations.*

Line 74. Could you provide the distance to the other strong beams too?

*The distance between the CS2 Point of Closest Approach (POCA) and the 1l strong beam is ~2200 metre, distance to the 3l strong beam is ~ 4500 metre whereas the distance to the 2l strong beam is ~1500 metre.*

*The following line has been added after Line 74:*

*"The strong beam 2l was ~1500 metre from the CS2 point of closest approach whereas the beam 1l and 3l were ~2200 metre and ~4500 metre away."*

Line 88. Provide which threshold is used by the re-tracker. Also, I thought it was known as the "CPOM" re-tracker? Maybe I'm wrong.

*The Cryosat-2 Baseline E Product Book mentions UCL sea ice retracker but in other studies such as Nab et al., (2023) the CPOM retracker is used. We assume both has been used interchangeably. The UCL sea ice/CPOM retracker determines the retracking point by applying a fixed percentrage threshold of 70% to the waveform's first maximum power return. The authors have included the threshold in Line 88.*

Line 105-110. Include abbreviations (Site 1 – S1; Site 1 Ridged; S1R etc.), as they are used later on in figures and text, but not typed out in the text.

*Noted and will be fixed in the revised manuscript.*

Line 135. Is Kwok et al. (2020) deriving it from total freeboard and sea ice freeboard? I believe it is the radar freeboard of CryoSat-2.

*Kwok et al., (2020) computed the ice freeboard from Cryosat-2 radar freeboard by accounting for the change in Ku-band velocity through the snow on sea ice using.*

*The line 135 has been revised and now reads: 'Kwok et al (2020) calculates snow depth (SD) as the difference between IS2-derived total freeboard (snow + ice) and CS2-derived radar freeboard (CS2), using the difference between the surface height and the instantaneous sea surface height interpolated from sea surface measurements from along-track leads to obtain the freeboards (Kwok et al., 2020; Ricker et al., 2014). The CS2 radar freeboard is also adjusted for reduced Ku-band speed to derive an accurate estimate of the sea ice freeboard.'*

Line 140. Ensure consistency with naming of h(IS2)/hIS2 and h(CS2)/hCS2 throughout text.

*Noted and will be fixed in the revised manuscript.*

Line 157. MSS is not mentioned in section 2.6. Please do.

*The line now reads: "However, the mean sea surface (MSS) ensuring that the IS2 ATL07 heights are referenced to the WGS84 ellipsoid. The MSS is calculated based on decadal averages and therefore are not representative of the variation of sea surface heights within the 77 minutes' interval between the IS2 and CS2 passes".*

Line 167. Do you have a reference for the Moran's I test?

*The following reference has been added to Line 167: Moran, P.A.P.: The interpretation of statistical maps, Journal of the Royal Statistical Society, 10, 243-251,https://www-jstor-org.uml.idm.oclc.org/stable/2983777.*

Figure 3. You mention co-registration of heights. What is meant here – how are you specifically co-registering the observations? Also, ensure consistency in the naming of h(CS2)/hi(CS2).

*The authors thank the author for pointing out that the details about the co-registration had not been mentioned in the text.*

*Line 171 now reads: 'After the spatial averaging of the ATL07 heights over 300 metre, the co-registration is conducted based on the distance to the closest CS2 Point of Closest approach. Therefore, each CS2 point is co-registered to the closest 300 metre ATL07 height segment.'*

Figure 4. I think that it is an interesting discussion and I like the idea of identifying comparable surfaces using Sentinel-1 backscatter. But I don't think the full picture of what is shown in the figure is being discussed in the text. You state that mean values are similar, therefore the assumption is that the same surfaces are observed – even when the distributions differ with a bi-modal distribution of IS2 and higher amount of high backscatter (between -16 and -14dB) observed than for CS2. Instead of doing a distribution-to-distribution comparison, could you instead do it **per smoothed, co-registered point and look at residuals of backscatter between the points**? To see how much they vary and at which locations they do differ. In addition, the standard deviation lines do not make sense to me – why are they not separated by the same distance around the mean values?

*The authors thank the reviewer for their valuable input and suggestion. As suggested, we computed the co-registered point wise difference between the retrieved IS2 and CS2 backscatter and tried to map the difference in backscatter (Figure R7). We see that the average difference in backscatter between the colocated points are within -+1 dB. The average difference in backscatter is 0.9 dB. Therefore, based on this analysis, we can assume that IS2 and CS2 may be seeing snow based on the Sentinel-1 backscatter and therefore may be colocated. The authors will include the findings of this analysis in the paper and also include the Figure G1 in the appendix.*

*The standard deviation of both the IS2 and CS2 varies by the same value about the mean backscatter. However, the visualization of the figure might have made this a bit harder to interpret. We will revise the figure to make the lines clearer.*

[Figure]

*Figure R7 Distribution of the difference in IS2 and CS2 Backscatter retrieved from IS2*

Line 207. Consider re-phrasing this, as it reads as if Farrell et al. (2020) computed surface roughness from standard deviation of 300-m segments, where it in fact was 25-km segments.

*The line now reads 'Surface roughness was calculated as the standard deviation of ATL07 sea ice height product following Farrell et al., (2020), however, instead calculating the roughness over 25 km, the regional differences in surface roughness were calculated over*

*300-meter length segments to maintain consistency with the spatially averaged ATL07 heights'.*

Line 213-214. I don't believe that Mallet et al. (2020) demonstrated that the use of fixed snow densities introduced significant biases in the snow depth retrieval, but rather significant biases in the sea ice thickness estimates. You are stating yourself, that the difference in snow density (to compute the refractive index) didn't make much of an impact on the snow depth (Line 219-220).

*The authors agree with the reviewer that the impact of snow densities on the retrieved snow depth retrievals have been demonstrated to be negligible. And yes, there was an error in interpreting Mallett et al., (2020) and therefore the line has been omitted from the revised manuscript.*

Figure 5. The text is quite small on the figure. I do wonder about distance sampled vs. number of samples at Site 2 (and for the others as well), since it was stated that you sampled every 5 m? Why is there such a difference in sample distance? Also, remove the table from figure and either incorporate into figure or include as separate table. Also, provide some comments/thoughts about the fact that you don't have an equal number of samples for each location.

*The authors thank the reviewer for identifying the difference in number of samples and distance sampled. The distance sampled was calculated based on the actual distance that was covered during the survey for each site. Although the sampling interval was intended to be 5 meter, the sampling interval varied between 2 to 3.8 meter due to variability in sampling by the magnaprobe user. We will include this limitation in execution of magnaprobe sampling in Line 131 which now reads 'The sampling interval was intended to be 5 m intervals to ensure spatial heterogeneity and avoid spatial autocorrelation of the sampled snow depth values following (Iacozza and Barber, 1999). However, the sampling interval was higher (2 to 3.8 metre) during the field sampling for all sites.'*

Figure 6. Consider changing the set-up of the subplots, so it is 1 row and 2 columns, as it will allow you to compare the salinity and density as a function on depth more easily. Also, remove table (incorporate in to figure perhaps?).

*Figure 6 has been adjusted to include 1 row and 2 columns and the table has been removed.*

Line 250. **I do not believe negative snow depths should be removed** – albeit not physically possible, they show the variation between IS2 and CS2 and provide insights in the differences between IS2 and CS2 too. In addition, it biases your statistics higher (which is already somewhat of an issue, since most of your C2I on average are smaller than the in-situ depths), so you should actually **be observing an even bigger difference**. Interesting that you do not see thicker snow than ~50 cm... I think you need a **figure of the actual co-registered ATL07 smoothed vs CS2 along-track data**, to truly see what is going on (as Figure 7 seems to show ATL07 in native resolution).

*The authors agree with the reviewer that it is important to include the negative snow depth values as part of the final computation of the snow depths. However, the impact of large*

*snow depths caused largely due to the noise of CS2 needs to be accounted for in the final snow depth estimation. As shown in Figure R8, there is a significant portion of negative snow depths lower than -10 cm which also corresponds with noise portions of the CS2 ellipsoid height which ultimately biases low the mean snow depth retrievals. These snow depths area attributed to the noise in CS2 data and would negatively bias the mean snow depths retrieved from Cryo2Ice across the track. Therefore, we disregard negative snow depths that are lower than 2 standard deviations from the mean and attribute this to the uncertainty due to the CS2 noise. Therefore, after adjusting for the outliers in the negative snow depths, we get a mean snow depth distribution of 7.52 cm. The new distribution of the Cryo2Ice snow depths are presented in Figure R9.*

*Subsequent results, discussions and conclusions have been adjusted to incorporate the new snow depth values.*

[Figure]

*Figure R8: Distribution of raw Cryo2Ice snow depth values including negative snow depth values*

[Figure]

*Figure R9: Cryo2Ice Snow Depths adjusted for negative snow depth values greater than 2 standard deviation from the mean dept*h

Figure 7. It is unclear in the legend, whether you're looking at ATL07 heights (in their native resolution) or your 300-m smoothed ATL07 data. Also, consider making the colors comparable across figures (sites have different colors to the vertical lines across figures). In addition, why are we seeing 5-km averages – they are not used nor mentioned in the text? **However, here you also see the variability of CS2 being more evident due to the majority of thin ice observations in IS2 data when smoothing at 5-km and CS2's noise**.

*Figure 7 has been revised to include the 300 meter averaged IS2 line in addition to the raw IS2 ATL07 heights.*

Line 264. What is meant by delineated with roughness? Please provide some more insights into this statement and reason for it.

*As mentioned in Section 2.5, Line 179, 'Roughness zones corresponding to each Site are defined as a portion of the CS2/IS2 track which had IS2 surface roughness (Section 2.6) within one standard deviation of the IS2 derived surface roughness directly adjacent to the*

*in-situ sampling site (Figure 1)'. Therefore, in order to define representative portions of the CS2/IS2 tracks corresponding with to the in-situ sites, the IS2 roughness was compared. The portion of the tracks which had roughness values within 1 standard deviation of the IS2 roughness closest to the Site was defined as a similar roughness site.*

Figure 8. Consider putting the latitude/snow depth plot on the y-axis of the image, so you can more easily compare it… Also, remember to include subplot numbers. In addition, perhaps I missed it, but how many C2I observations are used in this figure (and overall)? Somehow, these two plots make it look as if there are different number of observations shown.

*The total number of C2I observations is 214 and the number of observations is consistent with the map and graph. Since the map and the graph were presenting the same spatial distribution, we omitted the additional graph.*

Figure 10. Consider providing the specific information (bulk snow density) used to derive the snow depth of each site on the plot, for the reader to be reminded that for the site-specific calculations, different densities were used. Consider also providing information about the spread or similar in figure (or in other words, include Table 2 in Figure 10 as text).

*The snow bulk density for each site has been included in the legend of Figure 10. The information about the spread were referred to i.e. Table F1 instead of including in the figure since the diagram was becoming very busy.*

Section 4.3. I really appreciate this section and discussion, very nice!

*The authors thank the reviewer for their appreciation of the section.*

Line 330. "Radar heights can potentially be impacted by snow properties" – I would say that they certainly are!

The line now reads: *'While the IS2 green laser is mostly impacted by the air-snow interface conditions, CS2 radar waveforms interact with different layers of the snowpack and the dominant scattering horizon and subsequently radar heights are impacted by the snow properties'*

Table 1. Why are there two values at time in the mean snow depth column? Not fully clear.

*The two values are the range of means between different sites observed in these studies. The caption has been edited to make this clearer.*

Line 356. "Found bias of 2-5 cm, we can expect 15-40% systematic biases"… Maybe I missed it, but this seemed to be skipped relatively quickly. Could you provide more insights here? Also, what about the contribution of random uncertainties?

*The authors agree that this point needs further clarification. The systematic bias is ascertained based on the range of bias in snow depth compared with the mean snow depth retrieved.  The random biases have been quantified individually (surface roughness, tidal*

*variation etc) but it is difficult to ascertain the portion of bias caused by each uncertainty and therefore we are not mentioning a single range.*

---

## Author Response (AR2)

Dear author,

after a second round of review, both reviewers concluded that further changes are necessary, but these were considered minor. In addition to the reviewers' comments, I would have liked to see more emphasis placed on the fact that the study is based on a very limited dataset. This is not yet reflected in the title, which seems a bit too general. In addition, I am not yet convinced by the assumptions regarding the similarity of the Sentinel-1 radar backscatter, given the potential ambiguities in several parameters, including roughness and not just snow depth, and the relatively large spread of backscatter coefficients. The discussion of the semi-variogram results is criticized by both referees and more clarification is necessary.

Best regards

Lars Kaleschke

**Response to the Editor**

*The authors appreciate the editor's feedback on the paper and thank them for their detailed analysis of the paper. The authors agree that this study is based on a limited dataset needs to be stated explicitly and therefore the following lines in the abstract have been adjusted:*

*Line 21 now reads:*

*"In lieu of sea surface height estimates from leads, snow depths are retrieved using the absolute difference in surface heights (ellipsoidal heights) from ICESat-2 and Cryosat-2 after applying an ocean tide correction based on tidal gauges between satellite passes on 29th April 2022"*

*This ensures that it is clear to the reader that a single pair of passes were validated in the study.*

*An additional line has been included in the abstract which reads:*

*Line 30-31 now reads*

*'Moreover, the proposed methodology for getting snow depth over lead-less landfast sea ice needs to be validated using in-situ datasets in other landfast sea ice regions in the Arctic.'*

*This makes it clear that further studies are required to test the workflow proposed in this paper.*

*The authors would respectfully disagree with the Editor's suggestion on making modifications to the title. The suggested methodology is unique and has been tested using a representative dataset which included both satellite retrievals and in-situ field validation from lead-less, landfast ice in the Canadian Arctic Archipelago (CAA). The authors believe that the title is reflective of the new methodology proposed for obtaining snow depth over lead-less landfast ice using Cryo2ice. While the Canadian Arctic Archipelago (CAA) has been used as a study area, this methodology doesn't have any specific step that limits it to the CAA and therefore can be*

*implemented over lead-less landfast sea ice in other coastal regions where Cryo2Ice tracks coincidences and a tidal gauge station is present within a reasonable distance (~100 km).*

*The authors agree that the assumption of Sentinel-1 backscatter to compare snow distribution along the tracks were not sufficiently clarified and therefore have been revised. The Sentinel-1 backscatter is primarily considered sensitive to the surface roughness features over sea ice (Cafarella et al., 2019). However, surface roughness features are correlated with snow depth as demonstrated from the in-situ observations in this study i.e. larger snow depths in rougher areas and vice versa(Figure 1). Therefore, the general assumption is that the Sentinel-1 backscatter being sensitive to roughness will be sensitive to hundred meter scale sea ice features and therefore the broader-scale snow features. This assumption was not clearly stated previously but has been included in the revised paper. Additionally, a new Section 2.3 has been added to explain the Sentinel-1 data and processing steps used to obtain the backscatter.*

*Please note that The Sentinel-1 backscatter values are not considered for the snow depth estimations from Cryo2Ice and therefore the ambiguities with other parameters in addition to snow depth doesn't propagate uncertainties to the final snow depth calculation from Cryo2Ice. Therefore, the ambiguities in the Sentinel-1 backscatter due to roughness should not change the general trend in the snow distribution along the IS2 and CS2 tracks. The coincident Sentinel-1 imagery is primarily used as a diagnostic tool in order to identify which IS2 tracks should be compared to the CS2 track.*

*Lines 243 to 248 now reads:*

*"One of the critical assumptions is that IS2 and CS2 tracks are roughly coincident i.e. both tracks are measuring roughly the same snow despite their reference ground tracks being ~1.5 km apart.To test this assumption, Sentinel-1 SAR VH backscatter was characterized across both the IS2 and CS2 reference ground tracks. The sentinel-1 backscatter is sensitive to surface roughness which roughly corresponds to the snow depths along-track (Cafarella et a;., 2019). Therefore, the Sentinel-1 backscatter which is used to compare the backscatter profiles along IS2 and CS2 tracks to determine if they are similar and therefore are seeing similar snow depth distributions."*

*Across the variogram lags up to the length scale of the analysed orbit, the variance of the snow depths are pretty consistent and this is likely due to range noise in the 20 Hz CS2 observations. The methodology has therefore been modified to exclude the autocorrelation analysis. The authors do agree with the reviewers that the 1-km smoothing does provide valuable insigts. The authors have decided to include the 1-km smoothing based on the fact that the impact of CS2 noise may be reduced in addition to the native 300-m snow depth estimates from Cryo2Ice.*

**Reviewer 1**

Dear authors,

I'd like to thank you for the extensive and well-addressed responses to the reviewers. It's a treat to read well-thought-out comments and I'm happy to report that all my comments have been addressed – either completely or to some degree. I'm therefore happy to say that I support the publication of this manuscript after a few minor revisions.

In my previous review, my main issues revolved around the treatment of spatial scaling between ICESat-2 and CryoSat-2, and the alignment of the ellipsoidal heights of the identified CRYO2ICE observations correcting only for tidal adjustments. While the authors have done a great job investigating the impact and differences here, I'm not entirely convinced by some of the statements regarding the applicability of this treatment of spatial scaling. Hence, I'd like to ask the authors to consider the way the work is presented and open this up for discussion. I do believe that the methodology presented here is sound, but I worry that some of the statements are too rigorous and might be used "wrongly" in future studies when referenced. Thus, I urge the authors to consider, within the frame of these concerns, some of their statements and their choices for methodology highlighted below from the revised version of the manuscript or from the reviewer responses. Here, I believe discussing these points in the new section "4.3 Adjusting for the Difference in CS2 and IS2 Footprint" and in section "4.6 Surface Roughness and Cryo2Ice retrievals" would be enough.

*The authors would like to thank the Review for their in-depth analysis of the paper. The authors agree with most of the suggestions made by the reviewers and provide point by point responses to the raised points below.*

Major comments

Backscatter distributions

Line 230-232. I appreciate the inclusion of backscatter across all strong beams and the point-to-point comparison that you included in Appendix (Figure G1) – I actually think this is of more value to include in the manuscript as a main figure, than only Figure 4 alone – perhaps you could include both.

*The authors have adjusted Figure 4 to include both the backscatter plot as well as the point-to-point comparison.*

However, I also want to point out that Site 4 shows large variations in retrieved backscatter (more than -5 dB and 3 dB) suggesting some real differences here, which supports your conclusion that a lot of stuff has happened here (e.g., ridging, surface features not well captured in the coarser satellite sampling) – perhaps include this aspect in your roughness analysis. I would even consider making a backscatter histogram of the dB differences at each site (and include this along with Figure G1)…

*The authors are grateful to the reviewer for pointing out the large variation in backscatter in Site 4, this point has now been included in line 475-477 in the revised paper.*
*Lines 475-477 now reads:*

*'Significant variation in surface type in Site 4 is also evident from the large variation retrieved backscatter from Sentinel-1 (-5dB to 3dB)(Figure 4(b)) which was not very well represented from the snow depth estimations from Cryo2Ice.'*

*The backscatter histogram was not included for all the individual sites since the readers can pickup the spatial variation from the map in Figure 4(b) while the general backscatter trend along-track has already been provided in Figure 4(a)*

Spatial scales

I appreciate the authors work on evaluating the spatial autocorrelation with semi-variograms and the inclusion of an averaged product to 1 km. However, what I really want to open a discussion on is:

I. Do we believe that values of CryoSat-2 are "good" enough – in their own right – to track the surface so that one can derive meaningful statistics from observations in this native resolution, or to which extent (spatially) do we believe that the noise impacts?

II. Whether the semi-variogram you show truly presents an inflection point/convergence level at 1 km and some convergence after 300 m, when in fact you do not know what has occurred at shorter scales than 300 m due to the sampling of CryoSat-2? That is to say, you cannot really state that 300 m is where it convergences, when it is simply the first distance lag that you are presented with. Especially considering the large variability in your semi-variogram, does it truly reach an inflection point?

I'm mostly worried about discussion point II, is it appears that you state over landfast ice that the semi-variance after 300 m begins to become constant (but this is again your first lag-distance…) to indicate the spatial autocorrelation becoming negligible (and would be after 1 km). **So, while I do not believe that you need to change your methodology, since the inclusion of the 1-km smoothing provides some insights into the impact of choosing CS2's native resolution or not, I do believe you need to expand on this selection of footprints/smoothing scales and how applicable this is, in Section 4.3.**

  i.  *As noted in the Section 4.3, the native 300-meter product is helpful to estimate the snow depth closer to the small-scale roughness features such as ridges using Cryo2Ice. However, there is significant underestimation due to the introduction of range noise in CS2 20-Hz product which impacts the km scale snow depth retrievals. Therefore, the authors believe that the paper demonstrates the pros and cons of using the native and the smoothed resolutions and future studies can adopt either approach depending on the scale of the study. For example, for applications over few hundred meters, the native resolution may be used while smoothing needs to be applied for study areas over 1-km.*

  *Lines 527 to 529 now reads:*
  *Therefore, future studies should consider analyzing both the 300 meter resolution product and the 1-km averaged product in order to get both the meter scale snow depth variations from the 300 meter snow depths as well as the more representative snow depth distribution from the 1-km averaged snow depths.*

ii.

*Across the variogram lags up to the length scale of the analysed orbit, the variance of the snow depths are pretty consistent and this is likely due to range noise in the 20 Hz CS2 observations. The methodology has therefore been modified to exclude the autocorrelation analysis. The authors do agree with the reviewer that the 1-km smoothing does provide valuable insights and further discussion on the rationale needs to be included. The authors have decided to include the 1-km smoothing as an assumption in order to reduce the impact of range noise in the 20 Hz CS2 observations.*

Minor comments

While the discussion and putting in perspective to results of previous studies provided in the conclusion is interesting (!), it should not be mentioned – as a first time – in the conclusion, but rather during your discussion. I encourage you to include it further up.

*The discussion of the results from past studies have been shifted to Line 54 of the introduction instead of the conclusions.*

You included a new citation of a recent study (Fredensborg Hansen et al. 2024) and I believe you are citing it several times throughout the paper, but in different, mis-spelled, ways it appears. In line 50, you've written Freedensborg Hansne et al. (2024); in line 152 you've written Fosberg et al. (2024); line 464 it says Freesborgen Hansen et al (2024), and in line 296 and 474-475 you've correctly (according to your list of references) written Fredensborg Hansen et al. (2024). Please check and correct accordingly.

*The author apologizes for the sloppy error. The spellings have been corrected in the revised paper.*

Data (section 2). You've not included any sub-section on the Sentinel-1 data that you have used here to the backscatter comparison and the background of several figures? Please include this along with any pre-processing applied, and please include information about which polarizations you use (also, check for consistency that the same is used, e.g. Figure G1 and Figure 8 seem to either use different polarisations or colour-scales? Also, on these figures, please include the colourbar to show the range of dBs), and describe what you would expect over the landfast ice in terms of backscatter changes here.

*The authors thank the reviewer for their suggestion. The sentinel-1 backscatter information has been included in Section 2..3. The authors opted not to include the backscatter from the Sentinel-1 background imagery where the absolutely backscatter values are not relevant to the interpretation and therefore may potentially confuse the reader. The relevant backscatter analysis has been included in Figure 4.*

Also, some of the figures are still not of great quality and the text at times is too small to read (e.g., lat/lon on maps). I urge you to have another look at that!

*The authors appreciate the suggestion and the quality of the figures have been revised.*

Line 14-17. Does this in fact mean, that it is not only the first assessment of snow depth from CRYO2ICE over lead-less ice, but a first assessment of any dual-frequency snow depth over lead-less ice?

*The authors are not certain as previous airborne campaigns such as Operation Icebridge have only flown over landfast ice which focused on validating snow depth retrievals as well. Therefore, the authors would*

*limit the claim to Cryo2ice for this study as past dual-frequncy approaches may have been tested over landfast ice which the authors are not aware of.*

Line 20. "after applying an ocean tide correction (…)" - "after applying an ocean tide correction based on comparison with tide gauges (…)"

*The line has been revised based on the reviewer's suggestion.*

*Line 20 now reads:*

*In lieu of sea surface height estimates from leads, snow depths are retrieved using the absolute difference in surface heights (ellipsoidal heights) from ICESat-2 and Cryosat-2 after applying an ocean tide correction based on tidal gauges between satellite passes on 29th April 2022.*

Line 23. "significant" to "significantly"

*The line has been revised based on the reviewer's suggestion.*

*Line 23 now reads:*

*All four in-situ sites had snow with saline basal layers and different levels of roughness/ridging which significantly impacts the accuracy of the Cryo2Ice snow depth retrievals.*

Line 26. "attributing" to "attributes"

*The authors believe that "attributing" is grammatically correct here.*

Line 26-27. When you say surface roughness, do you mean snow surface roughness or ice surface roughness – or both?

*Primarily snow surface since the estimates are based on the standard deviation of the IS2 values.*

Line 37. "coincident" to "monthly composites of ": I'd be careful with the use of "coincident" here, as this is not really the case for the monthly estimates (there is some degree of spatial and temporal overlap, but in general they do not see the exact same ice).

*The line has been revised based on the reviewer' suggestion.*

Line 49. "A few hundred kilometers"? Fredensborg Hansen et al. (2024) shows overlap of more than 500 km, up to almost 1000 km for some tracks when consistent ICESat-2 and CryoSat-2 data was present. Perhaps change to "hundreds of kilometers". It sounds like there is little data available along transect, which is truly representative.

*The line has been revised based on the reviewer' suggestion.*

*Line 60 now reads:*

*This realignment means that once in every 19 CS2 (20 IS2) cycles, the two ground tracks nearly align for hundreds of kilometers over the Arctic providing new opportunities to improve and validate snow depths retrieved by combining laser and radar freeboards.*

Line 80. Remove "~" before "150 photons". I'm not aware of a change to the ATL07 methodology to not be exactly 150 photons, but maybe it has?

*The line has been revised based on the reviewer' suggestion.*

Line 82. "For this study, the ATL03"... I'm questioning this segment-length value based on a response to reviewer, where it was stated that you got 36 ATL07 segments within each 300-m segment, resulting in ~8.3 m segment-length. Is this the case for all your segments, or is this in fact an average of number of segments along your ~75-km track? That would suggest that you get exactly the same photons across the entire track, which seems unlikely. Could you clarify how this was computed?

*This is not the case for all the length segments, this is in fact the average of the number of segments along the ~75 km track.*

Line 103. "over sea ice in the SARIn mode" to "over sea ice floes in the SAR/SARIn modes". The re-tracking over leads is different! (although not relevant for your study, but important nonetheless).

*The line has been revised based on the reviewer's suggestion.*

*Line 114 now reads:*

*This fixed threshold retracker is used in the CS2 Baseline E level product over sea ice floes in the SAR/SARIn mode.*

Line 144. Could you include a reference to data/where you observed this showing the high pressure during this period?

*Reference to ECCC data has been included.*

Line 150. Sentence seems to stop abruptly. Remove "to" before references, perhaps?

*The line has been revised based on the reviewer' suggestion.*

Line 161. Re-check formula for refractive index! Should be ^1.5, right?

*The line has been revised based on the reviewer' suggestion.*

Line 179. How did you get this 6 cm difference? Tide gauge or from the tide models? It's not entirely clear from the updated text.

*From the tide gauge data, the line has been revised.*

*Lines 205-207 now reads:*

*'The tides varied over a range of ~ 6.0 cm in Dease Strait in between the two passes based on the tide gauge data, so it was crucial to check if the tidal corrections contained within the products accurately accounted for tide differences in the ~77 minutes between passes.'*

Line 186. You state here that it is a semi-variogram of the in-situ snow depths, but the semi-variogram that you have shown in response to reviewers was for the CRYO2ICE snow depths? If that is true, it is striking that they both reach an inflection point at ~1 km as you conclude, although I must say I don't fully believe the semi-variogram of CRYO2ICE at 1 km fully shows a deflection/convergence at 1 km due to the sampling of CryoSat-2 and the variability (as discussed in Major comments). If you have semi-variograms of both, I strongly encourage you to include it!

*The authors have decided to exclude the semi-variogram analysis based on feedback from reviewers.*

Line 231-232: I would probably change this sentence to reflect that in the majority of cases, the assumption is likely valid. But, in your point-to-point comparison you show some variability (which might be significant!).

*The line has been revised based on reviewer's feedback.*

Section 4.1 seems a bit short or insufficient, perhaps? Is there not more you can state when comparing with former studies? Perhaps on snow depth variability along the transect (e.g., your standard deviation compared with the ranges observed)? Expectations regarding the periods in question (beginning or end of late winter etc.)?

*The authors agree that the comparison with the past studies sub-section did seem to be short and therefore have been merged with the following Section 4.2 which discusses the snow depths from Cryo2Ice in further details.*

Figure 7. You did not update this to include the 1-km averages too? I strongly encourage you to do so.

*Figure 7 was included in Section 3.3 which included results from the native 300 meter resolution of IS2 and CS2 heights and snow depths. The 1-km averaged product was included as a part of the discussion section 4.2. Since the 1-km averaged heights are discussed later in the Discussion, Figure 7 only includes the original IS2 and CS2 heights along with the IS2 averaged over 300 meters.*

Line 312-315. You mention here the semi-variogram, but do not show it either as a main figure or in Appendix? I strongly recommend you do include it.

*The semi-variogram analysis has been removed based on feedback from reviewers.*

Line 374. What 100-km-averaged product?

*The line has been revised based on reviewer's feedback.*

Lines 376-387. Interesting discussion! It appears to me, that there is then some compensating biases between CryoSat-2 and ICESat-2 when smoothing to the 1 km length scales, since very smooth or very rough ice appears to have the largest differences, but transition zones (Site 1 and 2) match well. Interesting what could be driving this in the processing.

Figure 7 and 12. I do wonder whether instead of just a line, you could shade the "area" of equivalent coverage (the roughness segments/equivalent along-track coverage you show in the maps), to highlight the observations compared in the Site-specific plots.

*The roughness zones are computed based on point-wise information from IS2 and not 'area' of backscatter and therefore visualizing this as an area could mislead the readers to think that the retrievals were done over a set area instead of point-wise comparisons. This would also potentially bring in ambiguities related to the across-track variation in roughness for the labelled 'area'. Therefore, the authors believe that a line to represent the portion of the track that was defined as having similar roughness along-track is a better representation.*

Section 4.6. Since this is surface roughness from ICESat-2, it is essentially snow roughness – and here, it is based on the Gaussian Width of the photon distributions, if I am not mistaken. I think it would be

worth including a short paragraph on how well you believe this roughness parameter actually represents the roughness of the surface that you've encountered.

*The surface roughness in this study has been computed as the standard deviation ATL07 sea ice heights over 300 meter length segments.*

*Lien 478 to 482 now reads:*

*The surface roughness from IS2 computed and compared well to the roughness features picked up from the snow depth variations with higher roughness zones having higher snow depths from Cryo2Ice e.g. Site 4. However, the difference in spatial resolutions between IS2 and snow depths from Cryo2ice means that finer scale surface roughness features were missed by Cryo2Ice especially in the 1-km averaged snow depth product.*

Line 474. Could it also be because the negative snow depths in Fredensborg Hansen et al. (2024) are not at individual CS2 footprints, but rather at the 7-km averaged windows that you mention?

*The authors agree with the reviewers comment that the reason is definitely linked to the spatial scale considered.*

*Line 474 now reads:*

*We note that the number of negative freeboards (20%) is much larger than the 3% negative snow depths reported in Fredensborg Hansen et al., (2024) which we believe is mostly due to the fact that this study considers a single track averaging averaging over a 300 m and 1-km window compared to a 7-km window in the aforementioned study.*

Line 484. "(...) overestimation" to "(...) overestimation within vicinity of significantly ridged ice", or something similar to highlight that this was observed where you had large heterogeneity in the ice surface.

*The line has been revised based on reviewer's feedback.*

*Line 484 now reads:*

*We note that while Cryo2Ice generally underestimates snow depths by 2 to 4 cm compared to in-situ, the 1-km averaged snow depths also show the possibility of overestimation over significantly rough ice.*

Line 490. Remove "centimeter level" or "few centimeters".

*The line has been revised based on reviewer's feedback.*

Line 495. Remove "finer" after IS2. Not sure what is meant here. Perhaps "high-resolution" instead or something similarly, if that is was is hinted at?

*Additionally, there are uncertainties such as the use of a fixed threshold retracker in CS2 which is not tuned for the landfast sea ice and uncertainties associated with the IS2 fine- tracker that may also contribute significantly to the snow depth retrievals.*

**Reviewer 2**

2nd Review of "Snow Depth Estimation on Lead-less Landfast ice using Cryo2Ice satellite observations" by Saha et al.

All of my comments have been answered and I think the paper has been improved. But there are a few concerns left. Please find some additional comments below.

*The authors thank the reviewer for their responses and appreciation of the improvements made during the revision. The additional valid comments or concerns have been addressed below.*
Abstract, L25-29: I suggest to explicitly mention potential biases due to the CS2 main scattering horizon not being at the snow-ice interface (due to salinity, snow properties, etc.). Moreover, the choice of the retracker also affects the results as we do not know if it optimized for the given conditions.

*The authors agree that the point about the position of the CS2 main scattering horizon needs to be explicitly mentioned and this has been added in Line 26-27 of the abstract. This point has also been been introduced in Line 41-42 and finally discussed in Section 4.4. The authors do agree that the bias due to the choice of retracker has not been mentioned in the abstract and has now been added in Line 27.*

*Line 26 to 30 now reads:*

*The results suggest that it might be possible to estimate snow depth over landfast sea ice without leads. However, the observed biases of 2-4 cm likely stem from several factors: (1) discrepancies in sampling resolution between ICESat-2 and CryoSat-2, (2) the CryoSat-2 scattering horizon not aligning with the snow-ice interface due to snow salinity, density, and surface roughness, (3) the choice of retracker, and (4) potential errors in the altimeter's tidal corrections. Further investigation is needed to address these issues.*

Conclusions: See above, I think this should be also mentioned in the conclusions.

*The authors believe that the impact of snow geophysical properties on the position of the CS2 not being at the snow-ice interface has now been explicitly mentioned in Lines 531-532. Additionally, the impact of the choice of retrackers have also been mentioned in Line 498.*

*Line 531 now reads:*

*Snow geophysical properties, especially snow salinity in the deepest few centimeters of the snowpack, may impact the dominant scattering surface of the CS2 radar, resulting in the scattering surface shifted upwards into the snowpack, leading being further above the snow-ice interface which return and can leads to underestimation of the snow depths.*

*Line 541-543 now reads:*

*Additionally, there are uncertainties such as the use of a fixed threshold retracker in CS2 which is not tuned for the landfast sea ice and uncertainties associated with the IS2 fine- tracker that may also contribute significantly to the snow depth retrievals.*

Regarding the semi variogram analysis: I am still not convinced here. I think the noise, mainly in the CS-2 data is too high to derive values about the autocorrelation. Looking at Figure R5, there is no binned point indicating a drop for small distances, which is also because of the limited along-track resolution.

My interpretation of the Figure is that the 1 km distance that is identified in the paper, is mostly a result of the model being forced to meet the (0,0) point. Is this autocorrelation analysis really needed? I would rather state that something is based on a simple assumption rather than a method with potentially misleading results.

*The authors agree with the reviewer's suggestion that the autocorrelation analysis is indeed creating confusion and therefore can be excluded.*

Regarding negative snow depth: I agree that removing negative snow depths below the 2x standard deviations is a reasonable approach.

Figure 5: I suggest that you also show the negative snow depths in the histograms here, so it is consistent with the other figures.

*There were no negative snow depths measured from the in-situ snow depth measurements collected using a magnaprobe which was accurately calibrated. Therefore, negative snow depths don't make physical sense to be included as part of the in-situ snow depths shown on the histograms in Figure 5.*